# Smarter Sustainable Tourism: Data-Driven Multi-Perspective Parameter Discovery for Autonomous Design and Operations

Raniah Alsahafi [1], Ahmed Alzahrani [1] and Rashid Mehmood [2,*]

1   Department of Computer Science, Faculty of Computing and Information Technology, King Abdulaziz University, Jeddah 21589, Saudi Arabia
2   High-Performance Computing Center, King Abdulaziz University, Jeddah 21589, Saudi Arabia
*   Correspondence: rmehmood@kau.edu.sa

**Abstract:** Global natural and manmade events are exposing the fragility of the tourism industry and its impact on the global economy. Prior to the COVID-19 pandemic, tourism contributed 10.3% to the global GDP and employed 333 million people but saw a significant decline due to the pandemic. Sustainable and smart tourism requires collaboration from all stakeholders and a comprehensive understanding of global and local issues to drive responsible and innovative growth in the sector. This paper presents an approach for leveraging big data and deep learning to discover holistic, multi-perspective (e.g., local, cultural, national, and international), and objective information on a subject. Specifically, we develop a machine learning pipeline to extract parameters from the academic literature and public opinions on Twitter, providing a unique and comprehensive view of the industry from both academic and public perspectives. The academic-view dataset was created from the Scopus database and contains 156,759 research articles from 2000 to 2022, which were modelled to identify 33 distinct parameters in 4 categories: Tourism Types, Planning, Challenges, and Media and Technologies. A Twitter dataset of 485,813 tweets was collected over 18 months from March 2021 to August 2022 to showcase the public perception of tourism in Saudi Arabia, which was modelled to reveal 13 parameters categorized into two broader sets: Tourist Attractions and Tourism Services. The paper also presents a comprehensive knowledge structure and literature review of the tourism sector based on over 250 research articles. Discovering system parameters are required to embed autonomous capabilities in systems and for decision-making and problem-solving during system design and operations. The work presented in this paper has significant theoretical and practical implications in that it improves AI-based information discovery by extending the use of scientific literature, Twitter, and other sources for autonomous, holistic, dynamic optimizations of systems, promoting novel research in the tourism sector and contributing to the development of smart and sustainable societies.

**Keywords:** smart tourism; sustainable tourism; natural language processing (NLP); big data analytics; deep learning; machine learning; unsupervised learning; Bidirectional Encoder Representations from Transformers (BERT); literature review; smart societies

## 1. Introduction

Smart tourism [1,2] has developed as a sector of the smart societies concept [3,4], which integrates traditional tourism practices with the use of smart technologies to offer transformative and tailored solutions for the specific needs and requirements of travelers. However, the exploitative practices of capitalist approaches have had a detrimental effect on social, environmental, and economic sustainability. In response, sustainable tourism has emerged as an alternative approach that seeks to enhance both the tourism experience and promote longer-term sustainability [5–10].

## 1.1. Tourism in the Global Economy

Prior to COVID-19 (2019), the tourism sector, which was nearly three times larger than agriculture [11], amounted to 10.3% of the world's GDP, approximately USD 9.6 trillion [12]. In 2019, international visitors spent approximately USD 1.8 trillion, which represented 6.8% of total exports. In the same year, the sector accounted for one-quarter of all new jobs created globally and made up 10.3% of all jobs totaling 333 million [12].

COVID-19 severely impacted the tourism sector, reducing its contributions to the global GDP by nearly half in 2020, and subsequently recovering slowly in 2021 to a total global GDP contribution of USD 5.8 trillion [13]. In 2020, 62 million jobs were lost in the tourism sector, and the spending by domestic and international visitors declined by nearly 50% and 70%, respectively [12].

## 1.2. A Case Study on Tourism in Saudi Arabia: Diversification of Oil-Based Economy

For Saudi Arabia, in 2019, prior to COVID-19, travel and tourism amounted to 9.7% of the total national GDP, equaling 77.8 billion USD (see "Saudi Arabia 2022 Annual Research: Key highlights" at [12]). The sector contributed 1.58 million jobs, making up a 12.2% share of national jobs. The spending by domestic and international visitors in Saudi Arabia was 16.6 billion USD and 29.9 billion USD (10.4 percent of total exports), respectively.

There was a significant drop in all these economic figures due to the pandemic in Saudi Arabia. The 2021 economic figures for travel and tourism are as follows. The tourism sector was 6.5% (down from 9.7%) of the total national GDP, equaling 51.5 billion USD, and contributed 1.3 million jobs, making up a 10% share of national jobs. The spending by domestic and international visitors in Saudi Arabia was 16.2 billion USD and 6.1 billion USD (2.3% of total exports), respectively. The relatively low decline in domestic spending across the pandemic time is because a large part of local tourism is based on rural, natural, or religious sites and activities. Moreover, the restrictions on international travel and tourism were compensated by domestic travel. These economic figures for Saudi Arabia and other countries can be found in [12].

These numbers clearly show the vulnerability of the tourism sector and the national and global economies.

It is high time for new investors and other stakeholders in the tourism sector in Saudi Arabia, explained as follows. The Saudi economy has been heavily dependent on the oil industry. Saudi Arabia's vast oil reserves and position as the world's largest exporter of oil give it significant economic leverage in the global market. Its role in OPEC and close economic relationships with major oil-consuming countries allow it to influence the price and stability of the oil, making it a key player in the global geoeconomics of oil.

Saudi Arabia is diversifying its economy to reduce its reliance on oil, create new job opportunities, attract foreign investment, and prepare for the future in order to provide more stability, be more attractive to investors, and ensure its long-term prosperity. For instance, Saudi Arabia is investing in renewable energy as part of its efforts to diversify its energy mix and reduce reliance on oil. The goal is to produce 50% of the nation's electricity from clean energy sources by 2030, and the country has made significant progress in the development of solar and wind energy. It is also exploring other forms of renewable energy, such as geothermal and tidal energy, and has established initiatives and programs to support the development of renewables.

Tourism is an important target sector in Saudi Arabia's efforts to diversify its economy [14]. It is expected that the tourism sector will become the white oil, i.e., a major source of revenue for Saudi Arabia, similar to how (black) oil has traditionally been a major contributor to the country's economy [15]. Saudi Arabia, which is located in the Middle East, is a country known for its rich cultural history and significant religious sites such as Mecca and Medina. The government has been working to develop the country's tourism industry in recent years, with a focus on both domestic and international tourism. To support this growth, the government has implemented initiatives, for instance, the development of new infrastructure, such as hotels and airports, and the promotion of the country as a desti-

nation for events, conferences, and other business activities. There are opportunities for growth in various areas of tourism, including cultural and religious tourism, adventure and nature-based tourism, business and conference tourism, and medical tourism. In addition, Saudi Arabia is home to a number of cultural and natural attractions, including museums, galleries, the Asir National Park, and the Al-Hasa Oasis. Its cities, including Riyadh and Jeddah, are also popular tourist destinations. The planned city of NEOM is being developed as a hub for advanced technology and innovation, and there are several other planned tourism developments in Saudi Arabia, including Qiddiyah, Amaala, Trojena, Oxagon, The Line, Sindalah, and the Red Sea Project. Some of these are connected to NEOM city [16].

As Saudi Arabia and other Gulf countries compete in the tourism industry, they each offer unique cultural and natural attractions, modern infrastructure and facilities, and a convenient location for tourists from around the world. All of these recent developments and factors make researching the tourism sector in Saudi Arabia exciting and lucrative.

### 1.3. Technologies and Their Impact on Tourism

The tourism industry is being transformed by a number of key technologies, including virtual and augmented reality, artificial intelligence, machine and deep learning, the Internet of Things (IoT), mobile technologies, social media, big data and analytics, and blockchain. The IoT is expected to positively impact the tourism industry by providing timely or real-time interactions with tourists in areas such as transportation, attractions, tours, shopping, and hotels [17]. Artificial intelligence and machine learning are being employed to improve the customer experience through personalized recommendations and helping with trip planning, finding the nearest restaurants, suggesting road conditions, and many other applications for reservations, hotels, transport, and restaurants [18]. Virtual and augmented reality is being used to create immersive experiences for travelers [19]. Mobile technologies and social media have made it easier for travelers to book trips, find things to do, and connect with other travelers [20]. Big data and analytics are helping the industry to understand traveler behavior and preferences [21]. Blockchain has the potential to revolutionize the way the industry operates by enabling secure, transparent, and decentralized transactions [22]. These technologies are coming together to provide innovative capabilities and services including forecasting tourism demand [23], mining tourist locations, pathways, and travel itineraries [24], the visualization of tourist mobility activity patterns, identifying tourist congestion zones and tourist routes in specific areas [25], measuring tourist satisfaction [26], support for decision-making [27], identifying factors related to tourist satisfaction [28], integration of the Android platform and GPS services for guiding tourists to their preferred sites [29], augmented reality for historical tourism using mobile and smart devices [30,31], and more.

These technologies are all playing a significant role in transforming the tourism industry into smart tourism and shaping the way we travel in the future. Smart tourism [1,2] has developed as a sector of the smart societies concept [3,4], which integrates traditional tourism practices with the use of smart technology to offer tailored solutions for the specific needs and requirements of travelers.

### 1.4. The Need for Smart, Responsible, and Sustainable Tourism

Tourism is a multi-faceted industry that can bring economic and cultural benefits, but it can also pose challenges [32–34]. Environmental sustainability is a major concern, as the growth of tourism can put a strain on natural resources and ecosystems and contribute to climate change [35]. Overcrowding and over-tourism can lead to social and environmental impacts and overburden local infrastructure [36–40]. The economic impact can be positive, but can also lead to disruption and inequality [41]. Cultural sensitivity is also important as the influx of tourists can lead to cultural tension [37]. Health and safety risks, such as the spread of disease, crimes, and natural disasters, are also a concern [38,42]. Geopolitical dynamics and wars are also worsening the situation [43,44]. Many innovative and radical technologies are emerging frequently [45,46] while the existing technologies are evolving

at a fast pace, and these advancements and dynamics are making unforeseeable impacts on societies and hence on tourism, its nature, and requirements [30,31,47]. It is important for all stakeholders, such as governments, businesses, and local communities, to work together to develop sustainable and responsible tourism practices, a concept being pursued internationally under the sustainable tourism umbrella [5–10].

A holistic and dynamic (interactive, timely, or real-time) understanding of the sector and related global, regional, national, and cultural issues is needed to drive innovation and improvements toward smart and sustainable tourism [42]. Further details about the research gap can be found in Section 2, and a discussion on the novelty and utilization of this work can be found in Section 6.

### 1.5. This Work

This study uses our data-driven approach to model the tourism industry through the lens of both academics and the general public, using 156,000 research papers and 485,000 tweets. By combining advanced technologies such as deep learning and big data, we have developed a machine learning pipeline to extract parameters from both the academic literature and public opinions on Twitter. This approach gives a unique and comprehensive view of the tourism industry from two differing perspectives. These perspectives are not isolated and have some impact on each other, but they still have distinct and significant variations.

The aim of this study is to gain a comprehensive understanding of tourism, drive future research through cutting-edge technologies, and ultimately develop a theory and approach for smarter tourism that supports sustainable future societies. The paper presents a comprehensive knowledge structure and literature review of the tourism sector, drawing on more than 250 research articles.

The academic-view dataset was constructed using the Scopus database for the purpose of uncovering key elements of academia-oriented tourism. The dataset consisted of 156,759 English language research article abstracts and titles along with their keywords, covering the period from 2000 to 2022 (precisely 30 July 2022). By analyzing the academic dataset, we identified 33 distinct parameters relating to tourism and grouped them into four main categories, viz. Tourism Types, Tourism Planning, Tourism Challenges, and Media and Technologies.

For a period of 18 months from March 2021 to August 2022, we collected a Twitter dataset showcasing the public perception of the tourism sector in Saudi Arabia. The dataset comprised of 485,813 tweets and was limited to the region to focus on local tourism issues and compare them with international views. Our analysis revealed 13 parameters, which were then categorized into 2 broader set of parameters, Tourist Attractions and Tourism Services.

A software tool (see Section 3 for details) was developed for our data-driven approach to smart tourism, composed of four software modules: Data collection and storage, pre-processing, modelling and discovery, and validation, reporting, and visualization. The tool uses pre-trained BERT word-embeddings and UMAP for dimensionality reduction, HDBSCAN for clustering, and class-based TF-IDF scores to determine the importance of words in each cluster. The parameters are discovered from clusters and grouped into macro-parameters based on domain knowledge and are validated using internal and external methods. Visualization approaches, such as dataset histograms, taxonomies, and similarity matrices, are used to describe the data and clusters. Python packages such as Seaborn, Plotly, and Matplotlib were used to construct these visuals.

Figure 1 presents a multi-perspective taxonomy of the tourism sector, generated by our software tool. This taxonomy offers both academic and public perspectives, providing a comprehensive understanding of the industry. Additionally, it includes international and national or cultural (Saudi Arabia) perspectives, highlighting the diversity of the sector. The academic and international view provides a broad and in-depth analysis, covering 15 types of tourism, various planning dimensions, major challenges, and the impact of media

and technology. In contrast, the national and public perspective in Saudi Arabia focuses on services such as medical insurance and popular tourist attractions, including recent developments such as the trillion-dollar NEOM smart city, AlUla city, and seasonal festivals.

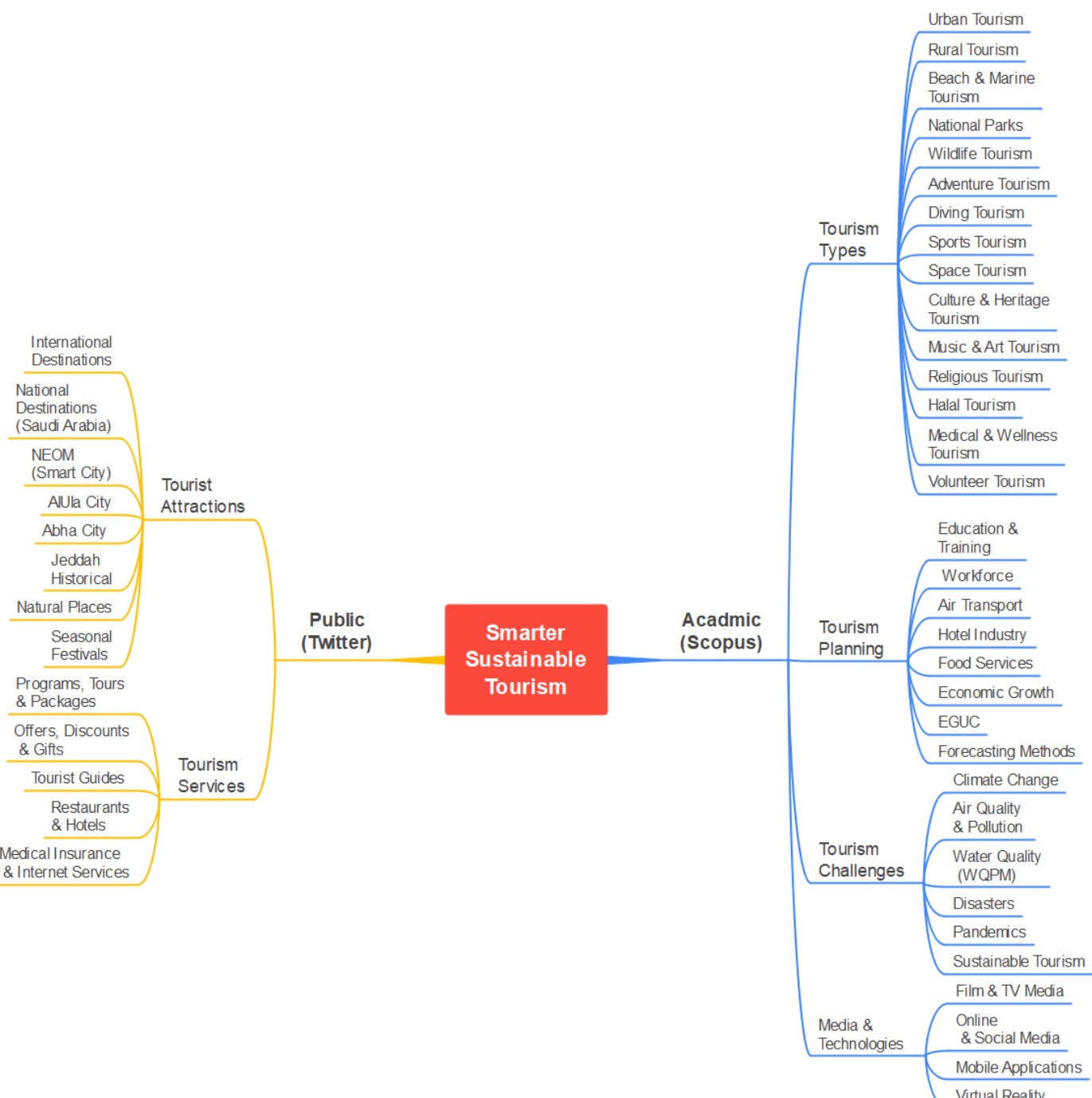

**Figure 1.** A multi-perspective taxonomy of smarter sustainable tourism.

The paper is organized as follows. Section 2 covers the related literature and identifies research gaps. Section 3 explains the research methodology and design of the proposed tool. Section 4 examines the parameters identified from an academic (and international) perspective using Scopus data. Section 5 examines the parameters identified from a public (and local or national) perspective using Twitter data. Section 6 presents a discussion. Section 7 concludes and suggests areas for future research.

## 2. State-of-the-Art and Research Gap

This section reviews works related to this paper. Our methodology that leverages deep learning has afforded us to conduct a comprehensive review of research on tourism using a dataset comprising 156,759 papers from the Scopus database. Additionally, we carried out an extensive review of research on the use of machine learning and data analytics in the tourism sector. No work directly linked to our study was found (i.e., a holistic, multi-perspective parameter discovery of the tourism sector using deep learning and big data analytics). However, to locate our study within the larger body of works on the use of machine learning and big data analytics in the tourism sector, we discuss related research from three fields. Firstly, we review research on the analysis of the academic literature (i.e., scientometric and bibliometric analysis works) in tourism. This is relevant because our work is based on the analysis of scientific literature on tourism. Subsequently, we review research in tourism on the application of artificial intelligence and machine learning for the analysis of digital and social media including Twitter. This is relevant because we analyze and extract parameters from Twitter data. Finally, we highlight the research gap in Section 2.3.

### 2.1. Data Analytics of Scientific Literature

Bibliometric and scientometric analysis of scientific literature has been used as an approach to analyze existing research in different areas such as finance [48], construction industry [49], transportation [50,51], smart homes [52], artificial intelligence [53,54], and others. Scientists have also employed scientometric analysis in tourism. For instance, Zach et al. [55] studied technology and innovation in the tourism sector using topic modelling. Loureiro et al. [19] used text mining to investigate augmented and virtual reality research in the tourism sector. Barrera-Barrera [56] used the LDA (Latent Dirichlet Allocation) algorithm for text mining research in tourism and hospitality with the aim to provide recommendations for selecting a suitable journal for submitting manuscripts using as criteria the manuscript topic, the journal scope, and the journal impact factor. Fang et al. [57] presented a scientometric investigation of the tourism literature with a focus on climate change using the CiteSpace tool. Chen and Zhou [58] used the CiteSpace tool and scientometric analysis to investigate people's motivation for travelling. Ribeiro et al. [59], using the VOSviewer tool, conducted a bibliometric analysis of the tourism literature with a focus on smart tourism. Baqeri et al. [60] employed scientometric analysis and the VOSviewer to investigate mining tourism with the aim of acquiring an understanding of the prospects of environmentally sustainable tourism involving the mining industry. It is clear from the literature that all the existing works have focused on specific topics in tourism. None of the works have attempted to use scientometrics to analyze the tourism literature in its broad sense and develop a comprehensive understanding of the tourism landscape.

### 2.2. Social Media Analytics

Data from digital and social media including Twitter has been widely used with textual analytics to study various phenomena and problems in different application domains such as in healthcare [61,62], disasters [63], smart homes [52], education [64], COVID-19-related studies [65,66], event detection [67,68], opinion mining about government services [69], city logistics [70,71], and transportation [51]. Likewise, in the tourism sector, several works involving textual analysis of Twitter data have been reported. Afzaal et al. [18] presented a classification of tourist sentiments from data in online reviews with the goal of extracting and categorizing users' favorable or negative sentiments about certain aspects. Ramanathan et al. [72] reported sentiment analysis of Twitter data with the aim of understanding tourists' views about tourism in Oman. Feizollah et al. [73] used Twitter data to investigate people's discussion topics and sentiments about halal tourism. Colladon et al. [24] reported a study based on the analysis of social networks and semantics of data from the travel forum TripAdvisor to forecast tourist arrivals in seven cities in Europe. Hasnat et al. [74] proposed a method using Twitter data for classifying tourists

and residents and determining the destination preferences of tourists. Obembe et al. [75] reported sentiment analysis of Twitter data to investigate the influence of social media and communications between stakeholders on the tourism sector during COVID-19. Vecchio et al. [1] showed how value can be created from social media data to improve a tourist destination.

The works that are specific to the analysis of tweets in the Arabic language for research related to tourism are but a few. Al Sari et al. [76] used Twitter, Snapchat, and Instagram data to study the performance of different machine learning algorithms for sentiment analysis of cruise experiences in Saudi Arabia. Alasmari and Abdelhafez [77] used machine learning to analyze Twitter data to investigate visitor sentiments about tourist destinations in Saudi Arabia. Al-Smadi et al. [78] reported sentiment analysis of hotel reviews in the Arabic language. Sayed et al. [79] compared the performance of machine learning algorithms for sentiment analysis of hotel reviews in Arabic. Al Omari et al. [80] proposed a Logistic Regression method for sentiment analysis of reviews of hotels, food, and shopping places.

None of the works on social media analytics in English, Arabic, or other languages have attempted to extract a holistic national understanding of tourism.

### 2.3. Research Gap and Novelty

The review of the literature we provided above establishes that the current research and development on smart tourism have primarily focused on investigating specific functions, activities, or aspects of tourism, such as the analysis of tourist sentiments, experiences, satisfaction levels, cruise experiences, food and accommodation experiences, travelling routes, forecasting tourism demand, and others.

While the existing literature is rich in breadth and depth, a comprehensive understanding of the tourism landscape is missing that is needed to better design and optimize the tourism sector, particularly because of the opportunities offered by the emerging technologies that allow sector-wide and intersystem, holistic optimizations. Moreover, none of the works on Twitter data analytics in English, Arabic, or other languages have analyzed tourism in its broad sense to develop a holistic national understanding of the tourism space.

Our work bridges these gaps and is novel due to its aim, design, methods, and results. None of the earlier works have been aimed at or designed to discover a multi-perspective, international, and national, holistic landscape of tourism. None of the earlier works have used a methodology and accordingly developed a tool that uses several cutting-edge deep learning and big data methods for the analysis, discovery, and visualization of information and knowledge for policymaking, design, and operations. An outcome of this work is deep learning-based discovery of information structure and taxonomy, design, and operations parameters, and a comprehensive review of the tourism sector literature.

## 3. Methodology and Design

In this section, the methodology and design of our proposed system are described in detail. The system architecture is displayed in Figure 2, which examines relevant topics that describe the tourism landscape, its various dimensions, its evolving nature, digital technologies and smart societies, tourist experiences, and tourism sustainability by using Scopus research articles. The software architecture is comprised of four components that will be discussed in the subsequent section. The overview of the system, including the main algorithm, is discussed in Sections 3.1–3.6, which describe the data collecting technique, data pre-processing, parameter modelling, parameter discovery and quantitative analysis, visualization, and validation, respectively.

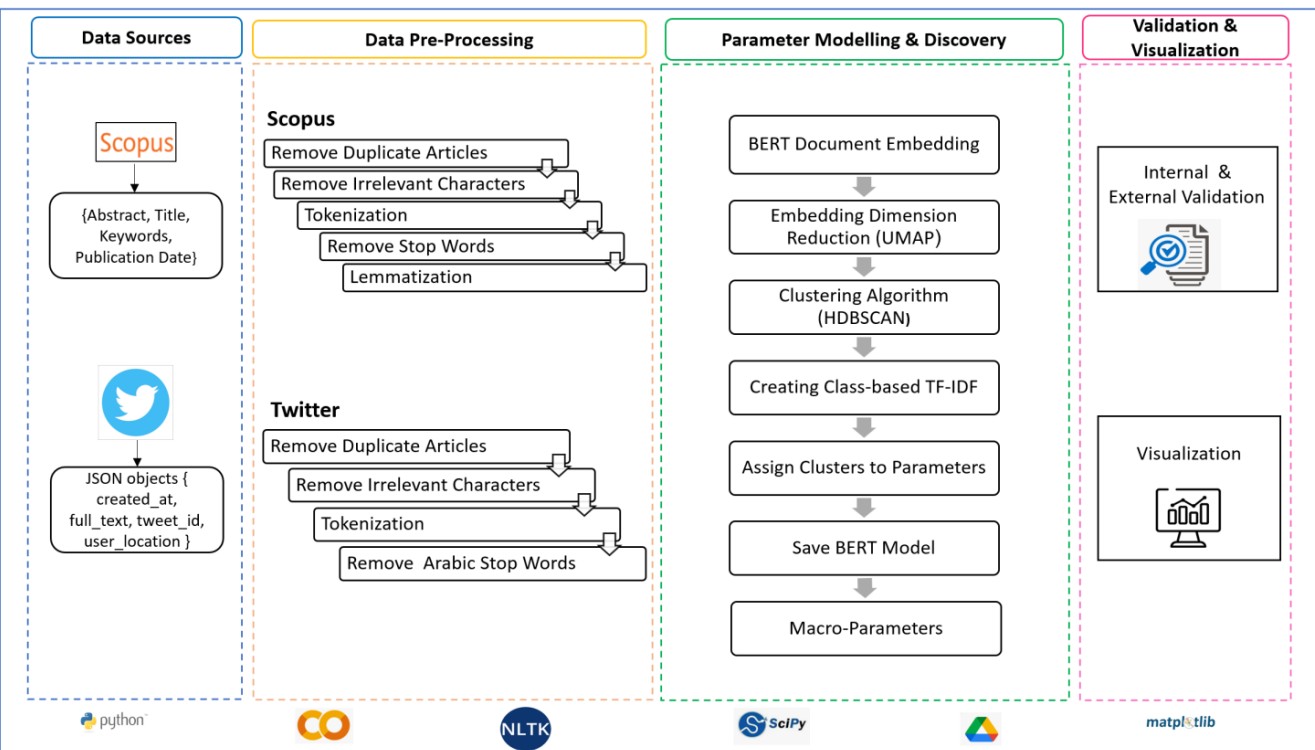

**Figure 2.** System architecture.

### 3.1. Methodology Overview

Algorithm 1 describes the fundamental steps at a high level of our proposed system. We performed a specific search query to obtain the data, then saved the results in CSV format. Then, we loaded the CSV file into the Panda data frame for the pre-processing step. During pre-processing, redundant articles, unnecessary characters, tokenization, stop words, and lemmatization are eliminated. Then, we utilized pre-trained BERT word embedding [81]. The essential part of the BERTopic model is the dimensionality reduction of the embeddings, therefore, we use UMAP. Subsequently, we applied the clustering technique using HDBSCAN to cluster the topics into groups of similar embeddings [82], and we calculate the class-based TF–IDF score to determine the significance of words within each cluster. In addition, we save the model and allocate each document to a cluster. Based on our domain expertise, we renamed the clusters as parameters and grouped these parameters into macro-parameters. The parameters and macro-parameters were thereafter displayed graphically. In addition, these parameters were validated using both external and internal validation.

---

**Algorithm 1** BERTopic Algorithm

---

Input: Scopus SearchQuery
Output: Topics with labeled parameters and visualization
1: CSV_file ← DataColletion (Scopus SearchQuery)
2: AbstractPapers_DF ← read_CSV (CSV_file)
3: PreProces_AbstractPapers ← DataPreprocessing (AbstractPapers_DF)
4: word_embedding ← CreateBERT_EmbeddingModel (PreProcess_AbstractPapers)
5: UMAP_Model ← DimensionReduction (word_embedding)
6: HDBSCAN_Model ← Clustering (UMAP_Model)
7: Caclcaulate_ClassTFIDF ← Clustering (HDBSCAN_Model)
8: BERT_Clusters ← ClusterRediction (Caclcaulate_ClassTFIDF)
9: Model ← LoadModel (BERTopicModel)
10: Parameters ← relabeled (BERT_Clusters)
11: figures ← Visualization (Parameters)

---

*3.2. Dataset Collection Method*

We utilized two types of data sources. The first one is the Scopus database, which we used to collect the academic literature, whereas the second data source is Twitter. We collected the pertinent papers from Scopus, the largest database of abstracts that includes expanded data and connected academic material from many different disciplines. The knowledge provided by Scopus helps researchers, librarians, research managers, and professionals make better decisions, take better actions, and produce better results. As such, we believe it offers an academic view of smart tourism. Articles might be produced to describe public perceptions and circumstances, yet these perceptions and expressions could be called academic because they are viewed and presented by academics. We used Twitter, a microblogging social media network, to determine the public's perspective on smart tourism. Twitter might contain tweets from governments, companies, and other interested individuals. Tweets from these and other parties might be utilized to understand other perspectives. We downloaded the Scopus academic literature from the Scopus website, then we used a CSV file to save them. Moreover, we used Twitter REST API to extract Arabic tweets then stored them in a CSV file.

3.2.1. Dataset (Scopus)

We collected 156,759 research documents by utilizing the "Tourism" keyword from five disciplines: Computer Science, Social Science, Environmental Science, Engineering, Business, and Management and Accounting. We limited our research documents type to articles, reviews, conference papers, conference reviews, book chapters, and books. Furthermore, we filtered the publication years as 2000–2022 and selected the English language. Faulty data such as duplicates and "no abstract available" were removed. The final data contained 100,244 articles.

Figure 3 illustrates the histogram of the Scopus abstract articles. The x-axis displays the number of academic research articles, while the y-axis displays the word count for each research article abstract. We note that many research article abstracts contained 250 words or more. The maximum number of words in the research abstract is 1500. The average abstract length of academic research abstracts is approximately182 words. Few research articles had more than 500 words.

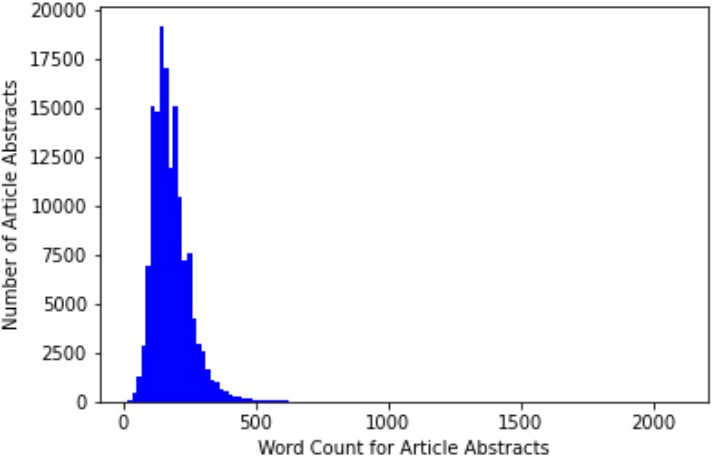

**Figure 3.** Histogram (Data Source: Scopus).

3.2.2. Dataset (Twitter)

The dataset collection phase used Twitter REST API to extract Arabic tweets. The Tweepy library and cursor python function were used to fetch tweets. The tweets were extracted using a search query for the keywords related to tourism, its types, and some Saudi cites, such as "السياحة السعودية" (Saudi Tourism), "السياحة" (Tourism), "السفر" (Travel), "السياحة الداخلية" (Local Tourism), "العلا" (AlUla), "نيوم" (Neom), and others. Moreover, we

used several tourism types such as, "السياحةالتاريخية" (Historical Tourism), "السياحةالبيئية"
(Environmental Tourism), "السياحة الدينية" (Religious Tourism), and others. Furthermore,
we utilized several hashtags related to the list of tourism activities and festivals in Saudi
Arabia and other hashtags for the official account of Saudi Tourism such as "موسم_الرياض#"
(# Riyad Season), "صيف_السعودية#" (#Saudi Summer), and others. We also collected data
using twitter official accounts that post about tourism in Saudi Arabia such as the Saudi
Tourism Authority (@SaudiTourism); (@VistSaudiAR), the official account of Saudi Tourism
Activates; (@GEA_SA), the official Twitter account of the general entertainment authority
in Saudi Arabia; and (@Tourism_news1), the account for publishing tourism news. Table 1
indicates a sample of the keywords, hashtags, and accounts that were used for scraping the
tweets. The total number of collected tweets is 485,813 tweets that were retrieved during
the period from March 2021 to August 2022. We saved the tweets in CSV format. The steps
of the data collection process are depicted in Algorithm 2. Firstly, we utilized a search
query to search for relevant tweets using the cursor Python function. We extracted the
tweet-created time, id, and text, then we saved them in a CSV file. Algorithm 2 illustrates
the steps of the data collection.

---

**Algorithm 2** Data Collection Algorithm

---

Input: Search_query {}
Output: Tweets_CVS File []
1: api ← connect_Twitter_API ()
2: Tweets_DF ← collect_tweetsTourism (created_at, tweet_id, tweet_text)
3: Tweets_CSV ← Tweets_DF

---

**Table 1.** The tourism keywords, hashtags, and Twitter accounts used to collect the dataset.

| Keywords |
| --- |
| السياحة، السفر، السياحة السعودية، السياحة الداخلية، سياح، مسافرين |
| Tourism, Travel, Saudi Tourism, Domestic Tourism, Tourists, Travelers |

| Hashtags |
| --- |
| #روح_السعودية، #السياحة_السعودية، #صيف_السعودية، #صيفنا_على_جوك، #السياحة_الداخلية، #موسم_الرياض، #اكتشف_العلا، #العلا، #نيوم |
| The Spirit of Saudi Arabia, Saudi Tourism, Saudi Summer, Our Summer, Internal Tourism, Riyadh Season, Discover AlUla, AlUla, NEOM |

| Accounts |
| --- |
| وزارة السياحة، أخبار السياحة، روح السعودية، الهيئة السعودية للسياحة، الهيئة العامة للترفيه |
| @Saudi MT, @Tourism news1, @VistSaudiAR, @SaudiTourism, @GEA_SA |

Figure 4 shows the histogram of Arabic tweets. The number of tweets is indicated
on the x-axis, while the word count of the tweet text is indicated on the y-axis. Tweets
are commonly less than 60 words long. There are relatively few tweets with more than
60 words in them. Only a small number of tweets have a maximum of 90 words, and a few
tweets have less than 10 words.

### 3.3. Data Preprocessing

During the data analysis, data pre-processing is an essential step after the collection
of data. Data pre-processing is conducted to improve the data analysis quality and ac-
curacy. This includes using several approaches to clean the data that has been collected.
Two different types of data sources were used in our research. English academic articles
were included in the Scopus dataset, whereas Arabic tweets were included in the Twitter
dataset. Twitter and Scopus cannot be pre-processed using the same technique because of
the varied languages utilized in both datasets. The following subsections describe the data
preprocessing processes for Scopus and Twitter individually.

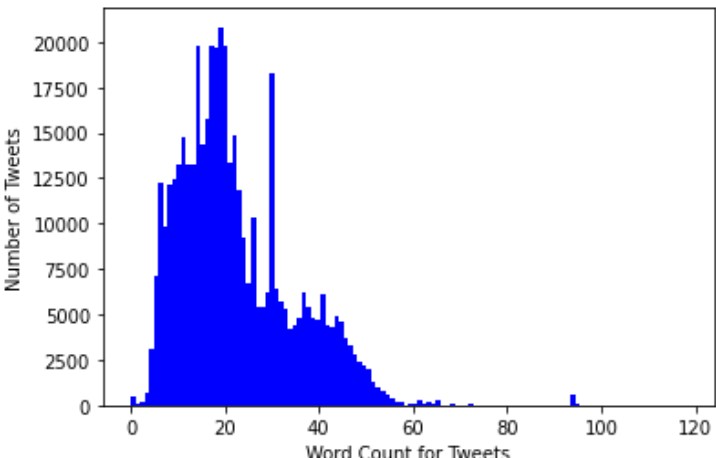

**Figure 4.** Histogram (data source: Twitter).

### 3.3.1. Pre-Processing (Scopus: English)

Among the pre-processing phases are the removal of duplicate articles, the removal of irrelevant characters, tokenization, and lemmatization using POS tags, and the removal of English stop words. Algorithm 3 illustrates the preprocessing steps as follows: First, we used a Python library (Panda) to read the CSV and saved it in the data frame. Second, we identified and removed redundant articles. Then, we deleted all unnecessary characters. In the fourth phase, the texts were tokenized using the Python package "genism", which contains the basic preprocess function. After that, the English stop words were removed using (NLTK) the Natural Language Toolkit list of predefined stop keywords. We initially developed the BERT parameter model and utilized the provided stop words list from NLTK for clustering. After obtaining the topics from the BERT model, we checked the associated keywords for each discovered parameter and investigated the irrelevant keywords that scored a high probability in the parameter. After some experimentation, we came up with a list of keywords and phrases that did not have a big impact on the parameters that were generated. Because of this, we removed certain keywords from the articles and added them to the list of stop words in our final model. Finally, the data were lemmatized using the WordNetLemmatizer. The cleaned abstracts were collected and saved in a CSV file for parameter modeling and discovery.

---

**Algorithm 3** Data Preprocessing (Scopus: English)

---

Input: abstract
Output: clean abstract
1: abstract _DF ← read_CSV (CSV_file)
2: RD_DF ← DuplicateRemoval(abstract_DF)
3: RNV_DF ← RemovaNoAbstractValue (RD_DF)
4: AIC_DF ← RremoveIrrelevantCharacters (RNV_DF)
5: token_DF ← tokenizer (AIC_DF)
6: RSW_DF ← removeStopWords(token_DF)
7: lemma_DF ← lemmatization (RSW_DF)
8: clean_DF ← cleanAbstract(lemma_DF)

---

### 3.3.2. Pre-Processing (Twitter: Arabic)

Algorithm 4 illustrates the preprocessing steps of Arabic Twitter datasets. The steps are explained in the following sections.

---

**Algorithm 4** Data Preprocessing (Twitter: Arabic)

---

Input: tweets_list; stopwords_List
Output: CleanTweets []
1: tweets_DF ← load(tweets_list)
2: tweets_RD ← RemoveDuplicate(tweets_DF)
3: tweets_RIC ← removeIrrelevantCharacters(tweets_RD)
4: tweetstokens ← tokenization(tweets_RIC)
5: tweets_normalize ← Normalization(tweets_tokes)
6: tweets_RS ← removeStopWords (tweets_normalize)
7: clean_Tweets [ ] ← cleantweets(tweets_RS)

---

Removing Duplicates Tweets and Irrelevant Characters

After collecting the tweets, we saved them in CSV files and loaded them into the Panda Data Frame format. We iterated the tweets and removed duplicate tweets to avoid the frequency of the same tweets. We defined the cleaning function that iterated the text of the tweet and removed numbers, the English alphabet, emails, URLs, extra whitespace, emojis, line breaks, and extra spaces. Moreover, all punctuations marks were removed because the majority of punctuation marks are misused. Therefore, we created a list of all punctuation containing periods (.), question marks (?), semi-colons (;), colons (:), commas (,), Arabic semicolons (؛), and Arabic question marks (؟), and we removed different types of brackets, mathematical symbols, and slashes. Furthermore, we deleted the hashtags and underscores to reduce the feature set's size and increase the value of information.

Tokenization

The tokenization process involved dividing text data into words (tokens). The NLTK library contains a module called tokenize (), which further classifies sentences into sub-sentences or words.

Normalization

The normalization process involves replacing letters that have multiple shapes with the same basic shape. For example, the Arabic letter Alif (أ), which is spelled Alif, has three shapes (آ ا إ), it normalized to (ا). In additionally, the letter Yaa (ي) is normalized to (ى) dotless Yaa and Taa Marboutah (ه ة) is normalized to dot Taa (ة).

Removing Stop-Words

In any language, there are many stop-words in textual data. Pre-processing involves removing these stop-words from textual data. Certain libraries are available in several languages for text preprocessing. An example is the Natural Language Toolkit (NLTK). It is used in a wide variety of languages, including Arabic. To remove the stop-words, we used the stop-words list provided by the NLTK library. In addition, we added a list of created stop-words that are usually used in dialectical Arabic, for example, "انتو", "ايش", "عشان", and "ليش".

*3.4. Parameter Modeling*

We used the BERT topic modeling approach in our system to cluster the data and identify parameters [83]. We created a word-embedding model before parameter modeling by utilizing BERT, a transformer-based method created by Google [81]. BERT is used to analyze text data and extract features, such as word and sentence embeddings. The sentence-transformer model known as "distilbert-base-nli-mean-tokens" was used in our work. Our documents were initially converted into numeric representations and generated dense vector representations for each document in the data corpus. The essential part of the BERTopic model is the dimensionality reduction of the embeddings; therefore, we used UMAP, which contains a considerable number of parameters. The n_neighbors and

n_components are the most important parameters. The parameter n_components refer to the dimensionality of the embedding. Depending on the circumstance, there are no direct approaches to select the optimum values. We tested the model several times, and we found that the n_neighbors = 20 and n_components = 7 produced the best result. Subsequently, we applied the clustering technique using HDBSCAN to cluster the topics into groups of related documents. Many parameters control the clustering technique, but min cluster size and min sample are the most significant HDBSCAN settings. The min cluster size specifies the minimum possible cluster size. The cluster size is controlled by the min sampling parameter. If the min sample is less than the min cluster size, the article will be merged into the same cluster. When the min sampling is large, more items are eliminated. The importance of terms for each parameter was also determined using a class-based TF-IDF score. The TF-IDF provides a way to evaluate the importance of words among texts by calculating a word's frequency in a specific document, as well as its importance throughout the whole corpus. However, if we regard each document inside a group as a distinct document prior to running TF-IDF, we will be able to evaluate the importance of each word within a cluster. This significance rating is known as the c-TF-IDF score. The parameter is more representative the more significant the words are inside a cluster. We may therefore obtain keyword-based descriptions for each parameter. Equation (1) is used to generate the c-TF-IDF score, where *f* is the word frequency for each class and *c* is determined and divided by the total number of words *w*. The overall frequency of words across all classes (*f*) is divided by the total number of unjoined texts (*d*) (*cc*).

$$\mathrm{c - TF - IDF} = \frac{f_c}{w_c} \times \log 10 \frac{d}{\sum_p^{cc} f_p} \tag{1}$$

Before the training model, it is challenging to predict how many topics will be produced once the topic model has been trained. We trained our documents using BERTopic, which produced results for several parameters. The nr_topic parameter determined the number of topics that will be reduced after training the topic model. It will reduce the number of topics to the specified nr_topics. We found the nr_topic = 45 is a reasonable number of topics. After that, we labeled all parameters and saved the model.

We re-labeled and aggregated the parameter—which was initially represented as an integer number—into macro-parameters using our domain expertise and quantitative analysis techniques. In the section below, we explain the quantitative analysis techniques.

*3.5. Parameter Discovery and Quantitative Analysis*

Understanding the topic model and how it works requires visualizing BERTopic and its variations. Users find it challenging to test their models since topic modeling may be a very subjective profession. An important step in solving this problem is to examine the issues and determine whether they make sense. Therefore, after our BERTopic model has been trained, we may iteratively go over hundreds of topics to truly understand the topics that were extracted. However, that takes a while and does not have a universal representation. However, visualizing the topics that were produced using quantitative analysis techniques might be a better approach. We identified the parameters and macro-parameters using domain expertise and quantitative analytic techniques, (i.e., Intertopic distance, keyword score, hierarchical clustering, and similarity matrix).

3.5.1. Term Score

A list of keywords (terms) for each parameter does not express the context of an associated parameter in the same manner. We need to determine the required number of keywords, as well as the beginning and ending points of essential keywords before we can identify a parameter. We sorted the keywords in decreasing order to visualize the c-TF-IDF score for each parameter [83]. The parameter may be identified significantly with the help of this word score visualization.

### 3.5.2. Inter-Topic Distance Map

The inter-Topic distance map generates a two-dimensional visualization of the parameters in an interactive Plotly graph. It is represented by parameter circles, the size of which is often related to the number of dictionary words required to describe that parameter. The circles are created using a MinMaxScaler algorithm. The parameters that are closer together share more words [83].

### 3.5.3. Hierarchical Clustering

The hierarchical clustering of parameters relies on cosine similarity matrices between parameter embeddings to connect parameters consistently [83]. Hierarchical clustering creates a unique cluster of hierarchical clusters by carefully matching clusters. All clusters are evaluated in all possible pairings beginning with the correlation matrix, and the pair with the greatest average inter-correlation inside the experimental cluster is selected as the new unique cluster.

### 3.5.4. Similarity Matrix

Applying cosine similarities to topic embeddings generates the similarity matrix [83]. The result will be a matrix displaying how closely related specific topics are to one another. The light green color displays the least similarity between parameters, while the dark blue color displays the largest similarity link.

### *3.6. Validation and Visualization*

The outcomes may be both internally and externally verified. Internal validation of parameters entails examining and analyzing the documents associated with the parameter. Documents in our study might be academic articles or tweets. In most of the documents in our collection, we explained how we judged the relationship between the documents and the parameters. External validation is measured by analyzing the parameters, keywords, and metrics of the two datasets. We applied a variety of visualization approaches for internal and external validation. Many visualization approaches are utilized to describe the datasets, clusters of documents, and identified parameters. Dataset histograms [84], taxonomies, hierarchical clustering [85], Intertopic Distance Maps [86], similarity matrices [87], Term Rank, Similarity Matrix [88], temporal progression charts [89], and word clouds are examples of these. Several Python packages, including Seaborn [90], Plotly [91], and Matplotlib [84], are used to construct these visuals.

### 4. Tourism Parameter Discovery for Tourism (Academia: Scopus)

In this section, we discuss the detected parameters by our BERT model obtained from the Scopus dataset.

Section 4.1 provides an overview of the parameters and macro-parameters. In Section 4.2, we provide quantitative analysis of the clustering characteristics and discovered parameters. In Section 4.3, we discuss each individual macro-parameter in detail and the temporal analysis of each macro-parameter.

### *4.1. Overview and Taxonomy (Academia: Scopus)*

We discovered a total of 45 clusters using the BERT modelling algorithm from the Scopus dataset. Our approach in this research was to cluster the data and then eliminate any unnecessary clusters. We omitted four clusters from the original clustering results as they were irrelevant to the work's subject. Four clusters (0, 24, 32, and 37) contain irrelevant words and the redundant keyword "tourism" due to the use of it in the research article. We grouped the 33 discovered parameters into four macros, termed macro-parameters, based on domain knowledge, the similarity matrix, hierarchical clustering, and other quantitative methods. The methodology and process used to discover parameters and group them into macro-parameters have already been described in Section 3.5.

Table 2 list the parameters and macro-parameters of the Tourism Scopus dataset. The parameters are categorized into four macro parameters, including Tourism Types, Tourism Planning, Tourism Challenges, and Media and Technologies (Column 1). Columns 2–4 list the parameters' names, their numbers, and the percentage of the articles for each parameter in the clustering model, accordingly. The fifth column represents the top 20 keywords related to each parameter. The BERT model identified 43.28% of documents as outlier clusters, so we ignored these clusters, and the remaining 56.7% of documents are listed in the fourth column.

**Table 2.** Macro-parameters and parameters for tourism (data source: Scopus).

| Macro | Parameters | No. | % | Keywords |
|---|---|---|---|---|
| Tourism Types | Urban Tourism | 18 | 0.34% | Urban, City, Urban Tourism, Culture, Space, Heritage, Branding, Tourism, Spatial, Social, City Branding, Economic, Historic, Destination, Sustainable, City Image, Policy, Urban, Planning, Design |
| | Rural Tourism | 1 | 1.31% | Rural, Rural Tourism, Farm, Tourism, Agricultural, Rural Area, Village, Community, Agritourism, Farmer, Landscape, Economic, Sustainable, Cultural, Traditional, Farming, Rural Community, Land, Product, Resource |
| | Beach & Marine Tourism | 22 | 0.31% | Coastal, Marine, Blue, Ocean, Area, Sea, Tourism, Maritime, Marine Tourism, Coastal Tourism, Resource, Economic, Ecosystem, Management, Island, Economy, Coastal Area, Sustainable, Coast, Coastal Zone |
| | | 31 | 0.24% | Beach, Coastal, Litter, Sand, Coast, Wave, Erosion, Management, Item, Area, Sediment, Marine, Sea, Along, Shoreline, Plastic, Natural, Recreational, User, Activity |
| | National Parks | 38 | 0.21% | Park, National Park, National, Visitor, Area, Protected, Management, Protected Area, Nature, Recreation, Ecotourism, Forest, Area, Cultural, Sustainable, Resource, Environmental, Community, Ecological, Landscape |
| | | 39 | 0.21% | Forest, Specie, Are, Land, Mountain, Plant, Bird, Vegetation, Landscape, Change, Cover, Fire, Park, Tree, Habitat, Management, Ecosystem, Population, Natural, Diversity |
| | Wildlife Tourism | 21 | 0.31% | Wildlife, Animal, Hunting, Specie, Human, Wild, Bear, Visitor, Population, Park, Experience, Habitat, Viewing, Area, Management, Encounter, Activity, Community, Disease, Protected |
| | | 29 | 0.25% | Wildlife, Animal, Elephant, Wildlife Tourism, Zoo, Specie, Human, Wild, Population, Bear, Tourism, Park, Hunting, Habitat, Visitor, Area, Management, Viewing, National Park, Encounter |
| | Adventure Tourism | 43 | 0.18% | Mountain, Adventure, Hiking, Trail, Alpine, Tourism, Area, Mountain Tourism, Landscape, Climbing, Hiker, Bike, Adventure Tourism, Destinations, Mountain Area, Biking, Activity, Management, Natural, Recreation |
| | Diving Tourism | 13 | 0.41% | Diving, Reef, Coral, Oil, Marine, Spill, Sea, Coral Reef, Coastal, Specie, Water, Island, Area, Site, Impact, Ecosystem, Coast, Environmental, Damage, Pollution |
| | Sports Tourism | 25 | 0.28% | Olympic, Sport, Game, Event, Olympic Game, Cup, Mega, Olympics, World Cup, Host, City, Legacy, Hosting, World, Host City, Sporting, Impact, Sport Tourism, Resident, Host |
| | | 33 | 0.25% | Olympic, Sport, Game, Event, Olympic Game, Cup, Mega, Olympics, World Cup, Host, City, Legacy, Hosting, World, Host City, Sporting, Impact, Sport Tourism, Resident, Host |
| | Space Tourism | 12 | 0.49% | Space, Flight, Launch, Commercial, Vehicle, Mission, Earth, Exploration, Station, Satellite, Private, International, Human, Technology, Cost, Design, Passenger, Law, Operation, Future |

**Table 2.** *Cont.*

| Macro | Parameters | No. | % | Keywords |
|---|---|---|---|---|
| | Culture & Heritage Tourism | 30 | 0.25% | Geological, Geotourism, Geopark, Geological Heritage, Heritage, Area, Site, Volcanic, Geomorphological, Unesco, Rock, Village, Building, Historical, Economic, Architectural, Resource, Global, Mining, Activity |
| | Music & Art Tourism | 16 | 0.37% | Festivals, Event, Music, Cultural, Visitor, Art, Community, Impact, Identity, Place, Organize, Culture, Economic, Group, Experience, Finding, City, Social, Held, Satisfaction |
| | Religious Tourism | 19 | 0.33% | Religious, Pilgrimage, Religious Tourism, Spiritual, Pilgrim, Religion, Sacred, Church, Tourism, Christian, Secular, Spiritual Tourism, Buddhist, Spirituality, Temple, Pilgrimage, Experience, Visitor, Heritage, Place |
| | Halal Tourism | 40 | 0.21% | Muslim, Islamic, Halal, Halal Tourism, Religious, Islam, Tourism, Destination, Compliant, Friendly, Country, Satisfaction, Religiosity, Non, Religious Tourism, Hotel, Attribute, Finding, Service, Saudi, Mosque, Pilgrimage, Pilgrim, Relationship |
| | Medical & Wellness Tourism | 6 | 1.03% | Medical, Medical Tourism, Health, Patient, Healthcare, Hospital, Wellness, Spa, Tourism, Health Tourism, Treatment, Service, Healthcare, Travel, Quality, Country, Eastern, Industry, Medicine, Factor |
| | Volunteer Tourism | 23 | 0.30% | Volunteer, Experience, Host, Organization, Community, Host Community, Participant, Motivation, Organization, Group, Project, International, Practice, Interview, Personal, Cultural, Global, Understand, Program, Motivation |
| Tourism Planning | Education & Training | 2 | 1.10% | Student, Tourism, Teacher, Work, University, Industry, Management, Tourism Student, International, Language, Graduate, Classroom, Educator, Program, Vocational, Degree, Training, Experience, Questionnaire, Degree |
| | Workforce | 17 | 0.34% | Employee, Job, Customer, Satisfaction, Work, Service, Organizational, Relationship, Performance, Workplace, Industry, Worker, Leadership, Organization, Manager, Engagement, Customer Satisfaction, Influence, Tourism, Working |
| | Air Transport | 36 | 0.23% | Airline, Airport, Air, Low Cost, Aviation, Passenger, Carrier, Cost, Transport, Low, Flight, Service, Travel, Industry, Demand, Route, Market, International, Traffic, Code |
| | Hotel Industry | 5 | 0.86% | Hotel, Customer, Hotel Industry, Industry, Employee, Performance, Guest, Manager, Satisfaction, Star, Management, Room, Green, Relationship, Hotel Manager, Star Hotel, Model, Organizational, Service Quality, Environmental |
| | Food Services | 3 | 1.1% | Food, Restaurant, Culinary, Food Tourism, Cuisine, Gastronomic, Gastronomy, Tourism, Experience, Product, Dining, Consumption, Culture, Customer, International, Safety, Service, Satisfaction, Industry, Quality |
| | Economic Growth | 15 | 0.38% | Growth, Economic Growth, Economic, Tourism, Model, Test, Demand, Country, Tourism Demand, International, Income, GDP, Arrival, Relationship, Result, Long, Long Run, Panel, Domestic, China |
| | | 10 | 0.56% | Tourism Industry, Growth, World, Growing, Economic, International, Market, Sector, Tourism Industry, Country, Largest, Travel, Economy, Fastest, Fastest Growing, Destination, Global, City, Increase |
| | EGUC | 27 | 0.27% | Poverty, Pro Poor, Tourism, Alleviation, Poverty Alleviation, Pro, Community, Income, Poverty Reduction, Economic, Country, Growth, Sector, Government, Benefit, Low, Development, Rural, Policy, Household |
| | Forecasting Methods | 14 | 0.42% | Forecasting, Model, Algorithm, Google, Forecast, Ontology, Forecast, Time Series, Fuzzy, Recommendation, Prediction, Neural Network, Performance, Arrival, Accurate, Combination, Result, Error, Corpus, Machine |

**Table 2.** *Cont*.

| Macro | Parameters | No. | % | Keywords |
|---|---|---|---|---|
| Tourism Challenges | Climate Change | 44 | 0.17% | Climate, Weather, Temperature, Thermal, Summer, Climatic, Dog, Meteorological, Outdoor, Season, Tourism, Weather Condition, Day, Warm, Comfortable, Humidity, Tourism Climate, Winter, Daily, Warm, Humidity, Comfortable, Pet, Seasonal |
| | | 20 | 0.32% | Ski, Winter, Climate, Resort, Climate Change, Ice, Mountain, Sport, Alpine, Area, Season, Condition, Adaption, Impact, High, Temperature, Cover, Destination, Alp |
| | Air Quality & Pollution | 26 | 0.28% | Carbon, Emission, Low carbon, climate, carbon emission, tourism, $CO_2$, Environmental, co2 emission, Consumption, Climate, Change, Growth, Policy, Country, Travel, Economic, Greenhouse, Industry, Air |
| | WQPM | 4 | 0.92% | Water, Lake, River, Water Quality, Groundwater, Water Resource, Quality, Waste, Water, Basin, Pollution, Drought, Irrigation, Management, Water Supply, Reservoir, Ecosystem, Sediment, Damage, High, Environmental |
| | Disasters | 35 | 0.23% | Tsunami, Disaster, Earthquake, Damage, Recovery, Flood, Area, Landslide, Natural Disaster, Hazard, Coastal, Natural, Affected, Tourism, Vulnerability, Indian Ocean, Resilience, Building, Impact, Affected |
| | | 28 | 0.26% | Death, War, Site, Memorial, Literary, Tourism, Heritage, Museum, Memory, History, Visitor, Place, World War, World, Visit, Military, Book, Experience, Narrative, Right |
| | Pandemics | 9 | 0.60% | COVID-19, 19, Pandemic, 19 Pandemic, Crisis, Disease, Industry, Health, Travel, Impact, Outbreak, Risk, Sector, 2020, Affected, Business Measure Global, Countries, Policy |
| | Sustainable Tourism | 42 | 0.18% | Sustainable, Sustainable Tourism, Tourism, Community, Environmental, Sustainability, Amazon, Manager, Heritage, Cultural, Policy, Impact, Attitude, Destination, Economic, Protection, Visitor, Ecotourism, Industry, Management |
| Media & Technologies | Film & TV Media | 41 | 0.20% | Film, Film Tourism, Movie, Television, Induced Tourism, Tv, Drama, Tourism, Popular, Screen, Video, Production, Cinema, Celebrity, Film Television, Audience, Fan, Cultural, Destination Image, Travel |
| | Online & Social Media | 11 | 0.56% | Online, Internet, Website, Travel, Social Media, Golf, Facebook, User, Hotel, Customer, Marketing, Consumer, Twitter, Review, Online Travel, Booking, Instagram, Travel Agency, Sentiment, Purchase |
| | Mobile Applications | 8 | 0.66% | Mobile, Smart, Smart Tourism, Smartphone, Mobile Device, Phone, City, Tourism, Technology, Mobile Application, Information, User, Smartphones, Location, Context, Experience, Map, GPS, Platform, Design |
| | Virtual Reality | 34 | 0.23% | 3d, Algorithm, Detection, Building, Virtual, Computer, Digital, Simulation, Software, Modelling, Sensor, Machine, Optimization, Laser, Visualization, Reconstruction, High, City, Network, Photogrammetry |
| | | 7 | 0.67% | Destination, Intention, Satisfaction, Vr, Image, Model, Relationship, Perceived, Effect, Service, Result, Finding, Destination Image, Influence, Customer, Quality, Factor, Emotion, Decision, Implication, Implication |

The taxonomy was extracted for the 33 parameters that were detected by our BERT model (See Figure 5). The taxonomy was created from Table 2, and it indicates the parameters, their macro-parameters, and some keywords related to the parameters. The first-level branches indicate the macro parameters, the second-level branches indicate the discovered parameters, and the third-level branches show the most representative keywords associated with each parameter.

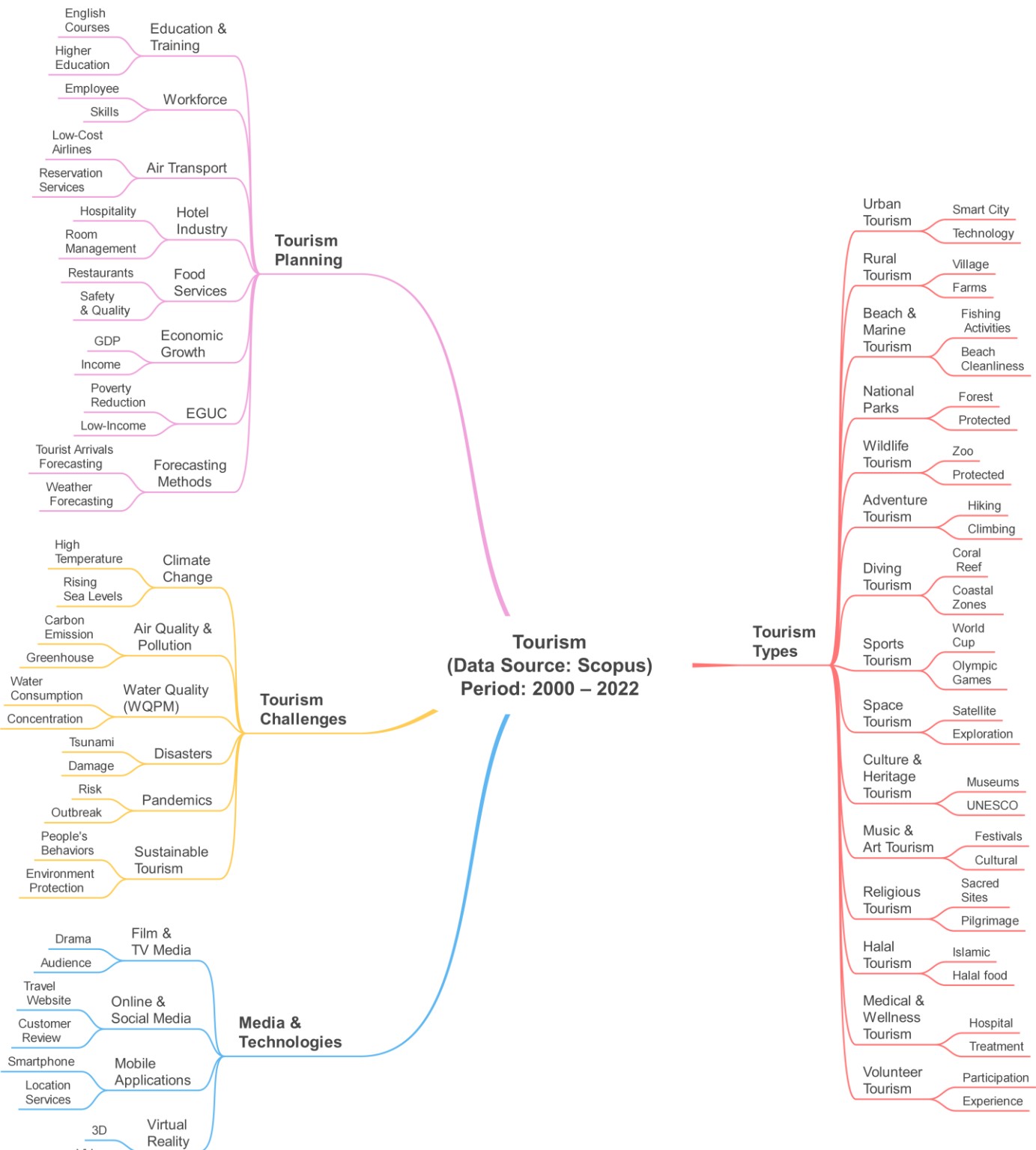

**Figure 5.** A taxonomy of tourism extracted from the Scopus dataset.

*4.2. Quantitative Analysis (Scopus)*

This section presents the quantitative analysis including the term score, word score, Intertopic distance map, hierarchical clustering, and similarity matrix. A set of keywords are used to represent each parameter; however, not all of them accurately define it. Figure 6 demonstrates that the first ten to thirteen terms in each topic's ranking accurately reflect

the topic. Since the probability of all other words are so close to one another, their ordering is essentially meaningless. Therefore, in order to label the parameter, we focused on its top ten to thirteen terms.

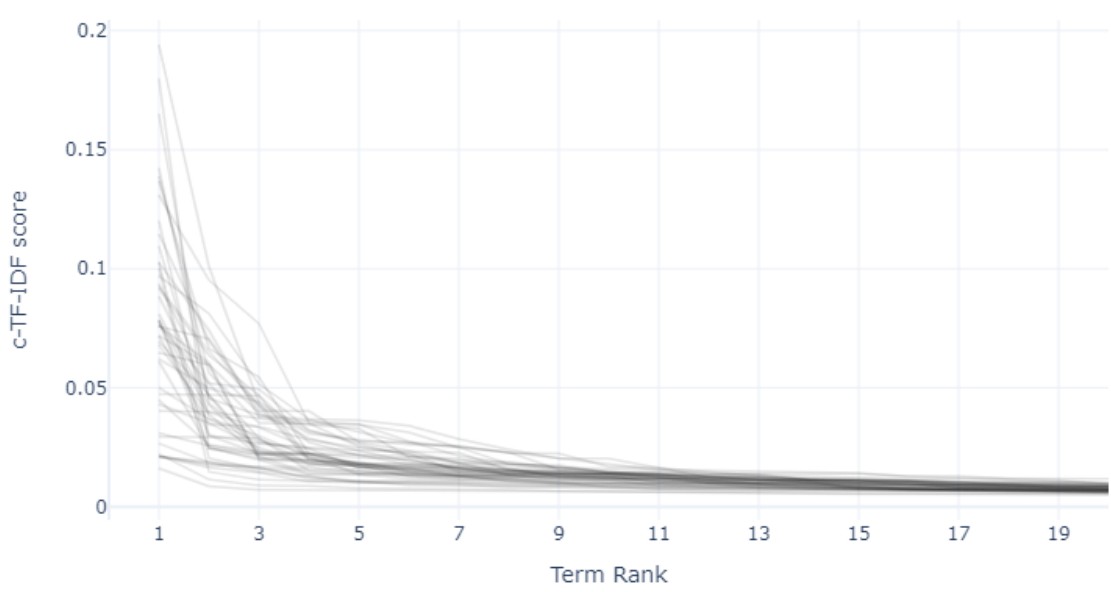

**Figure 6.** Term rank (data source: Scopus).

Figures 7 and 8 visualize the top 10 keywords for each parameter (see Section 3.5). The importance score, or c-TF-IDF, is used to order the keywords. There are 33 subfigures, and in each subfigure, the horizontal line shows the importance score, and the vertical line shows the parameter keywords.

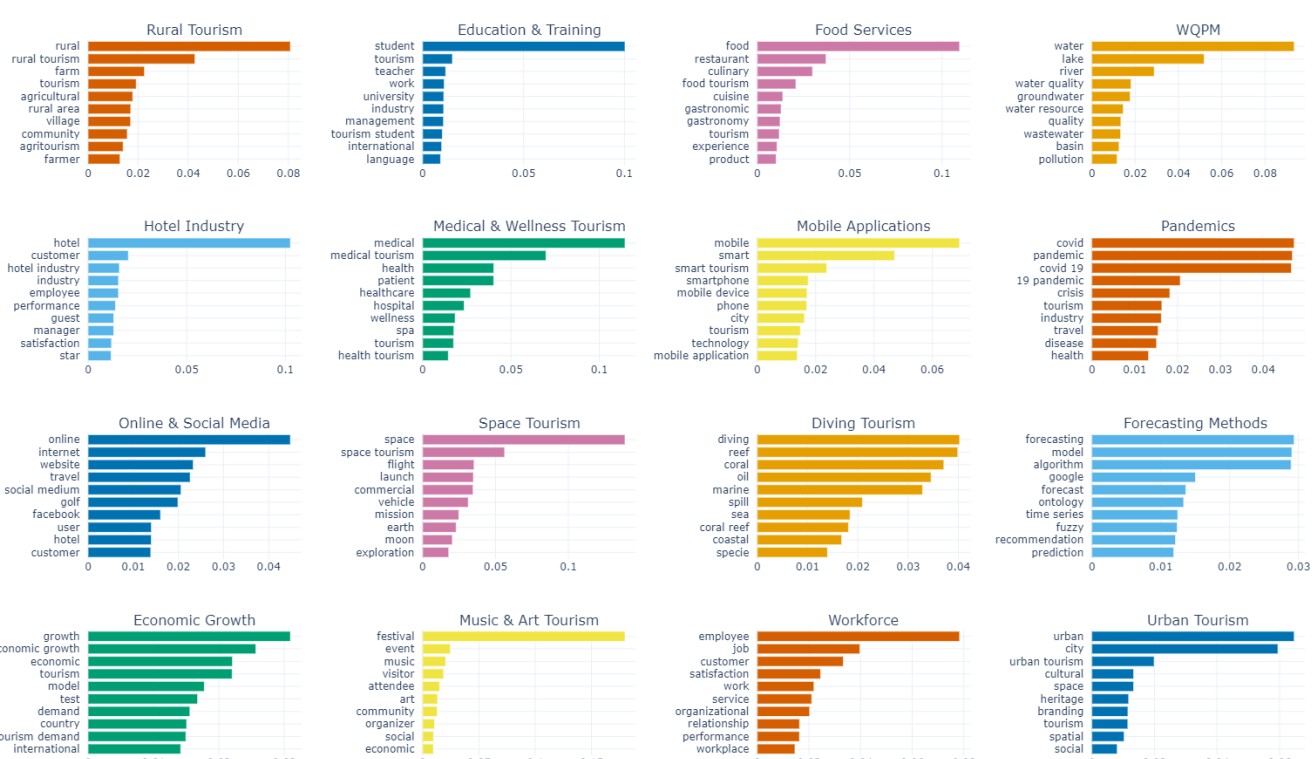

**Figure 7.** Smart tourism parameters with keywords' c-TF-IDF scores (data source: Scopus) (Part A).

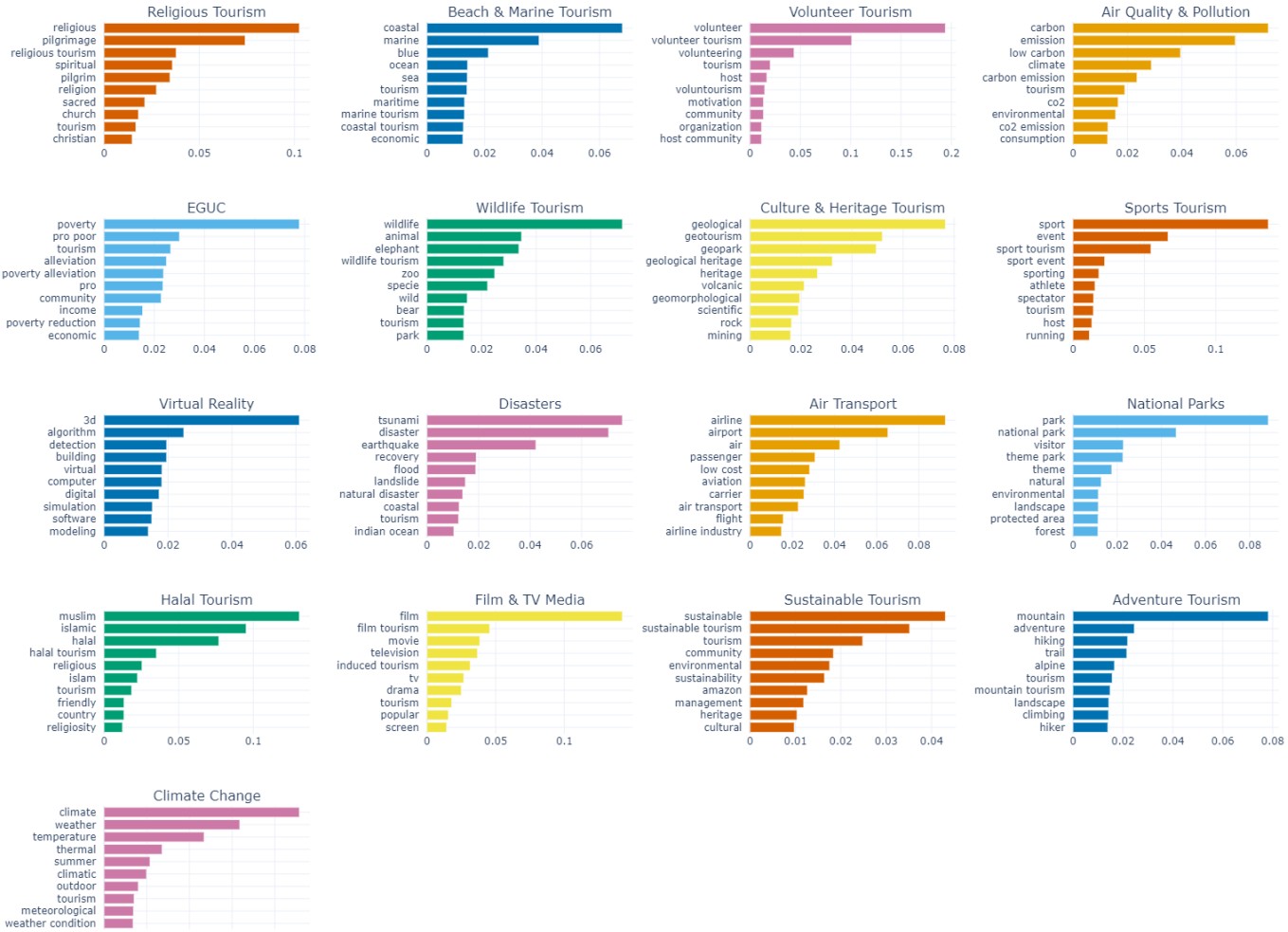

**Figure 8.** Smart tourism parameters with keywords' c-TF-IDF scores (data source: Scopus) (Part B).

Figure 9 indicates the hierarchical clustering of the 33 parameters and systematically pairs them based on the cosine similarity matrix (see Section 3.5). We noticed that clusters 1, 42, 18, 43, 38, and 29 created a unique cluster that we labelled the Tourism Types parameter.

Figure 10 shows the similarity matrix among the parameters (see Section 3.5). The dark blue shows the highest similarity between parameters, whereas the light green color shows the least similarity. For example, we note there is a dark blue color between cluster 20 (Climate Change) and cluster 43 (Adventure Tourism), which indicates a high similarity score because climate change is one of the most important factors that could affect adventure tourism.

Figure 11 shows the Intertopic Distance Map based on a multidimensional scale. The figure clearly identified four macro-clusters, where three clusters are clearly identified on the right side, and the lower-left side represents one cluster.

### 4.3. Tourism Types

This macro-parameter captures various dimensions of the tourism landscape in relation to the following parameters: Urban Tourism, Rural Tourism, Beach and Marine Tourism, Natural Parks, Wildlife Tourism, Adventure Tourism, Diving Tourism, Sports Tourism, Space Tourism, Cultural and Heritage Tourism, Music and Art Tourism, Religious Tourism, Halal Tourism, Medical and Wellness Tourism, and Volunteer Tourism.

### 4.3.1. Urban Tourism

This parameter captures tourism in urban areas and its link with technologies and smart cities. The keywords in this parameter include Urban, City, Urban Tourism, Culture, Space, Heritage, Branding, Tourism, Spatial, Social, City Branding, Economic, Historic, Destination, Sustainable, City Image, Policy, Urban, Planning, Design, Creative, and Study. The dimensions and research areas related to this parameter include the mobility management of tourist flows [92], providing better tourist services [93], improving street parking systems [94], green tourism [95], strategies to develop cities with sustainable hospitality [36], and the causes and effects of over-tourism in urban cities [36].

### 4.3.2. Rural Tourism

The keywords that were detected by our model include Rural, Farm, Rural Area, Agricultural, Community, Area, Farmer, Village, Economic, Sustainable, Activity, Rural Community, Landscape, Resource, Social, Farming, Cultural, Business, Traditional, and Result. The Rural Tourism parameter concerns tourism involving farms, rural areas, and agricultural activities bringing many benefits to the local communities. These benefits include the economic growth of local communities [96] and sustainable development of rural areas [97]. Rural Tourism is often seen as a good option for rural development and poverty reduction [98] and a major source for creating job opportunities in rural areas [99]. For instance, farmers may benefit from tourism by offering accommodation or selling farm products to tourists [100].

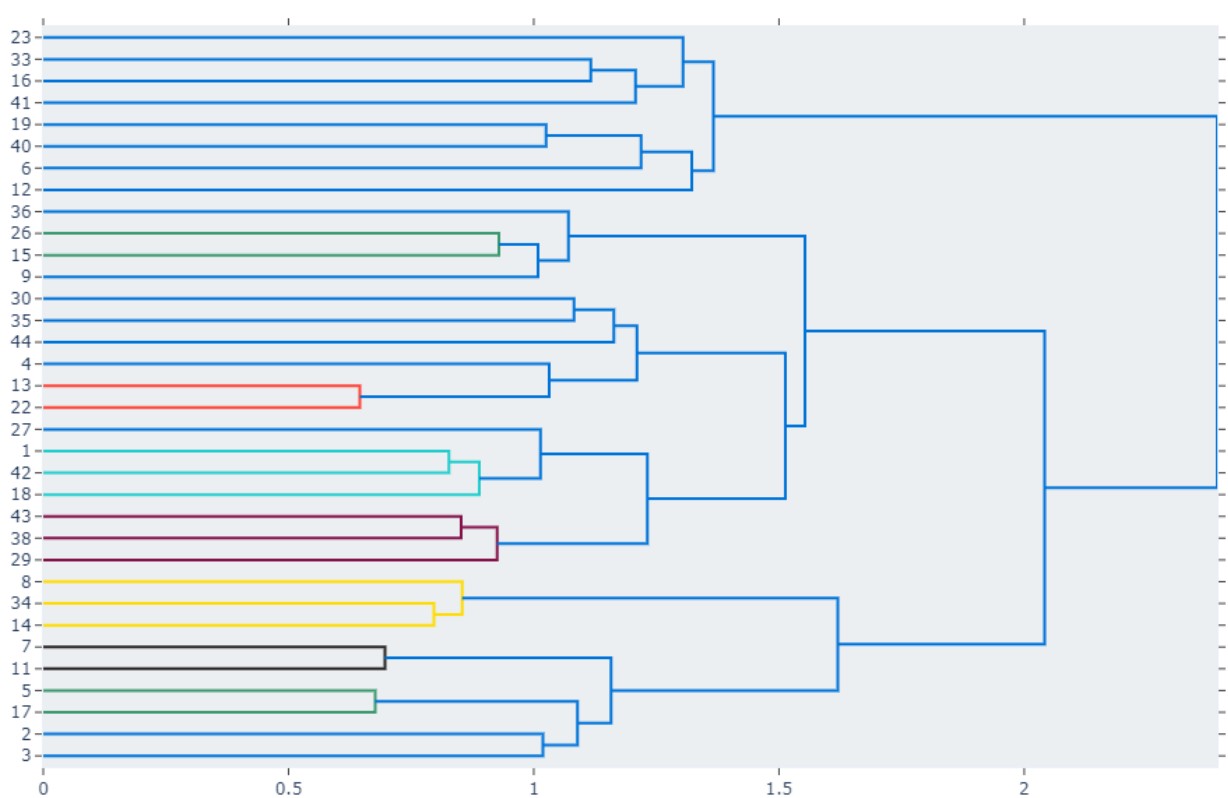

**Figure 9.** Hierarchical clusters (data source: Scopus).

### 4.3.3. Beach and Marine Tourism

The Beach and Marine Tourism parameter is created by merging two clusters (numbers 22 and 31) because the clusters include keywords pointing to similar subjects. It contains the following keywords: Marine, Island, Sea, Site, Coastal, Ecosystem, Contact, Activity, Area, Management, Recreational, Water, Specie, Impact, Damage, Ocean, Change, Cover, Environmental, and Environment. Beach and Marine are the main assets of tourist destinations. This parameter captures the issues related to the development, protection, and environmental sustainability of beaches and coastal and marine areas of the county. The parameter dimensions include developing a beach management strategy [101], ensuring the cleanliness of beach environments [102], protecting tourism infrastructure in beach zones [103], reducing environmental pollution, which can affect beach tourism experiences [104], ensuring the safety of water transportation to support the growth of marine tourism [105], and managing fishing activities in the maritime protected area [106].

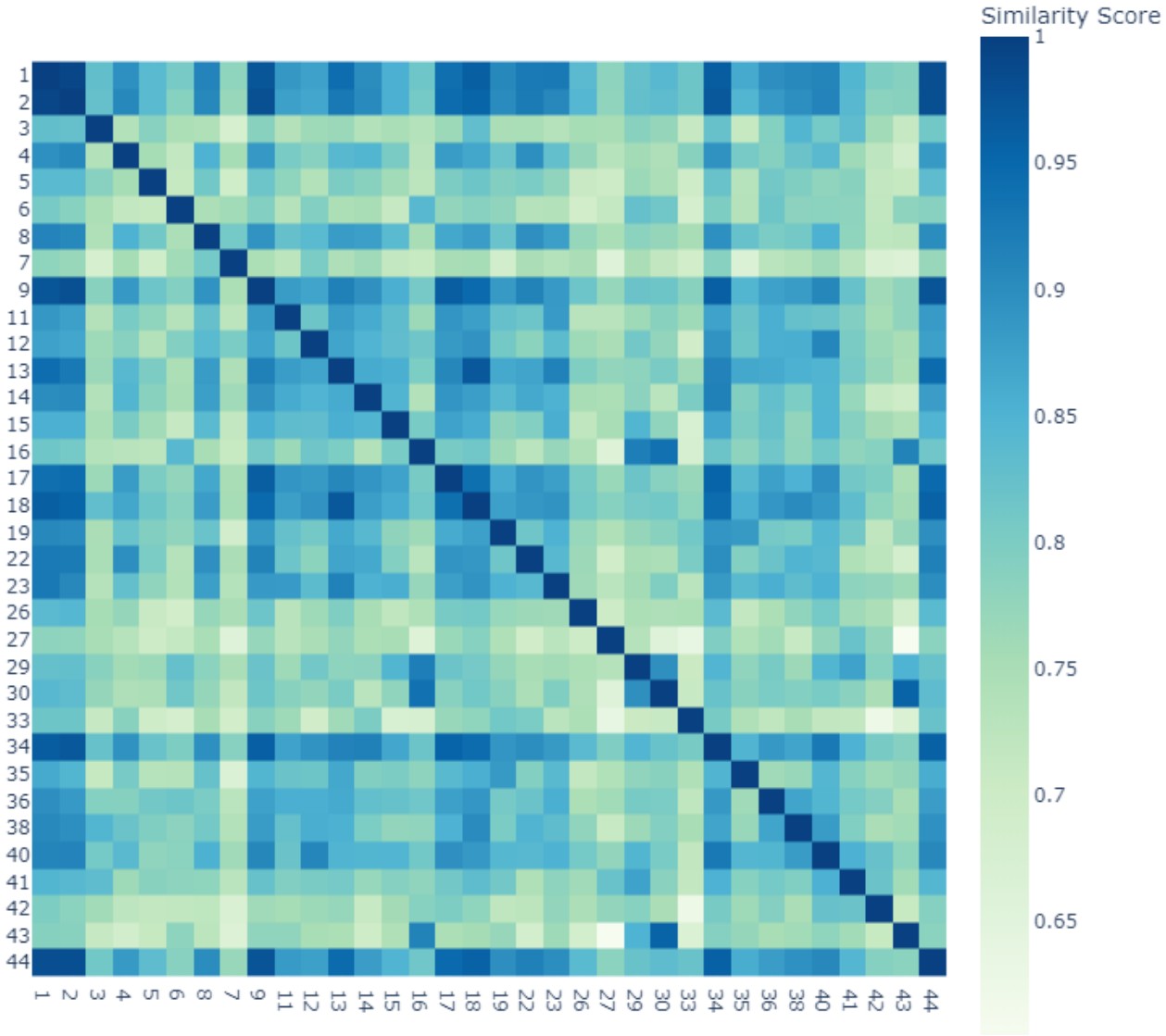

**Figure 10.** Similarity matrix (data source: Scopus).

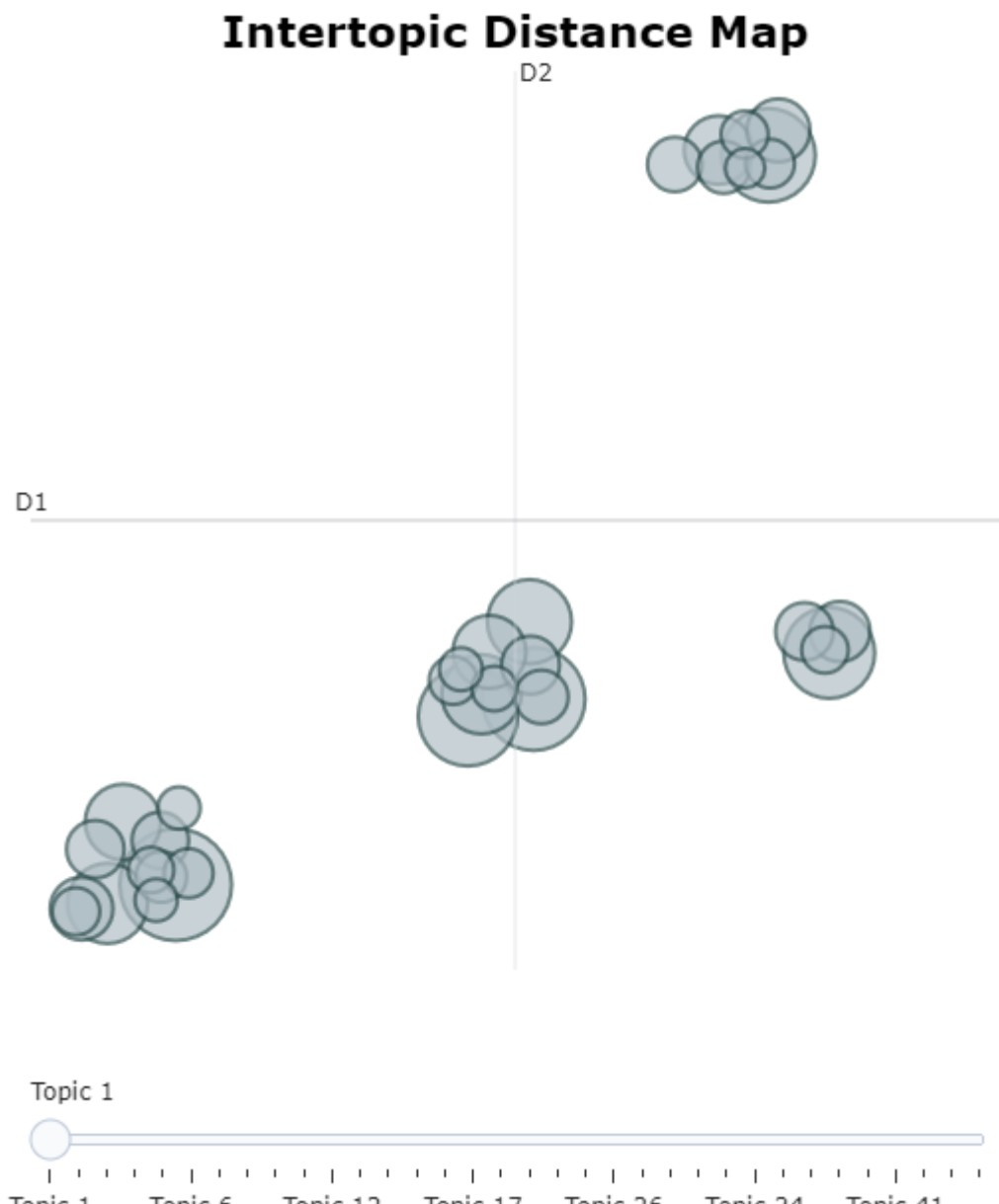

**Figure 11.** Intertopic distance map (data source: Scopus).

### 4.3.4. National Parks

The National Parks parameter explores the following keywords: Park, National Park, National, Visitor, Area, Protected, Management, Protected Area, Nature, Recreation, Ecotourism, Natural, Forest, Area, Cultural, Sustainable, Resource, Environmental, Community, Ecological, Landscape, and Village. This parameter emphasizes the importance of National Parks as tourist attractions designed to protect the natural environment from ecosystem proliferation. For example, Emphandhu et al. [107] applied a set of strategic policies to enhance local communities and National Parks. National Parks are very valuable natural areas and have the potential to attract a large number of visitors. Rogowski et al. [108] managed car traffic in National Parks to avoid over-tourism. Liu et al. [109] investigated the impact of climate change on the number of visitors in National Parks as climate change has a significant influence on rainfall and climate temperature. Kariyawasam et al. [110] investigated the role of National Parks in ensuring socioeconomic sustainability. Puustinen et al. [111] discussed how the number of visits is connected to the qualities of National Park, the leisure services within it, and tourism services in the neighboring areas.

### 4.3.5. Wildlife Tourism

This parameter concerns Wildlife Tourism activities in their natural habitats and tourists' experience. It is created by merging clusters 21 and 29. It contains the following keywords: Animal, Elephant, Wildlife Tourism, Zoo, Specie, Human, Population, Bear, Habitat, Visitor, Area, Management, National Park, Whale, Boat, Marine, Tour, Industry, Behavior, and Interaction. We discovered several topics related to this parameter including developing interpretations at wildlife tourist destinations such as zoos, aquariums, and others to increase visitors' understanding and appreciation for natural resources, as well as to make them aware of the effects that human activity has on animal populations [112], investigating the potential impacts of post-visit action tools such as printed handouts and email updates on the development of sustainable behavior following a wildlife tourism experience at a zoo [113], examining animal cloning and how it can help future tourist efforts [114], visiting and playing with animals in their natural environment such as watching whale and shark cage diving [115], and enhancing tourists' Wildlife Tourism experiences to achieve environmental sustainability by analyzing online reviews of wildlife tourism [116]. For instance, Flower et al. [117] looked at the types of tourists most likely to visit various elephant tourism locations and examined visitors' sentiments before and after their visits to try to raise public knowledge of the problems within elephant tourism sites in order to encourage positive attitude and behavior change.

### 4.3.6. Adventure Tourism

The Adventure Tourism parameter relates to travelling that involves a certain level of adventure and risk and may require specific skills such as climbing and physical exertion. It contains the following keywords: Mountain, Adventure, Hiking, Trail, Alpine, Tourism, Area, Mountain Tourism, Landscape, Climbing, Hiker, Bike, Adventure Tourism, Destinations, Mountain Area, Biking, Activity, Management, Natural, and Recreation. The dimensions in this parameter include creating an Adventure Tourism package, assisting in the improvement of product offerings in areas with natural resources [118], analyzing how visitors and tour guides behave in terms of sustainability during Adventure Tourism trips in natural places [119], evaluating the level of customer satisfaction of an adventure trip provider [120], risk management in Adventure Tourism by assisting tourism organizers and adventure tour guides in identifying risk and developing risk mitigation and risk reduction techniques [121], and investigating how Adventure Tourism may help the destination's tourist growth [122].

### 4.3.7. Diving Tourism

Diving Tourism relates to diving activities, coral reefs, and associated environmental impacts. It is represented by the following keywords: Diving, Reef, Coral, Oil, Marine, Spill, Sea, Coral Reef, Coastal, Specie, Water, Island, Area, Site, Impact, Ecosystem, Coast, Environmental, Damage, and Pollution. We discovered several dimensions and topics by reviewing academic documents that belong to this parameter including, among others, managing awareness of the significance of external risks to local diving operators [123], constructing artificial reefs to preserve coral reefs from several threats such as climate change, heavy human activity, and commercial usage, providing new tourist destinations that change diving and diving experiences [124], developing scuba diving guide features to ensure diver safety and environmental impact considerations associated with this activity [125], supporting the socioeconomic enhancement of coastal zones and islands by developing maritime infrastructure such as diving parks [126], and educating scuba divers to achieve environmental goals for coral reef protection [127].

### 4.3.8. Sports Tourism

Sports Tourism captures important dimensions of tourism related to sports events. It is represented by the following keywords: Sport, Event, Game, Mega, Sporting, Host, City, Legacy, Impact, Hosting, World, Destination, Economic, Participant, Image, Fan,

Social, International, Resident, and Group. Studies under this parameter discussed several important factors related to Sports Tourism including how sports events enhance tourism by bringing foreign visitors to a city to encourage economic growth [128], how foreign world cup tourists generate significantly higher returns compared to foreign leisure tourists [129], the importance of cycling and racing sports that bring positive participation in upcoming sport events [130], building a recommendation system to notify users who are more likely to attend an upcoming active sporting event [131], and using smartphone technology to enhance tourists' viewing experiences and provide information about sport events and enable facility booking, buying of tickets, and more [132].

### 4.3.9. Space Tourism

The Space Tourism parameter is described by the following keywords: Space, Flight, Launch, Commercial, Vehicle, Mission, Earth, Exploration, Station, Satellite, Private, International, Human, Technology, Cost, Design, Passenger, Law, Operation, and Future. We discovered several topics linked to this parameter by reviewing the Scopus academic articles that related to it. For example, Aravindhan et al. [133] discussed the importance of medical examinations during space travel for successful survival and the difficult performance of human exploration throughout space flight. It consists of training, selecting spaceflight, and recovery from several side effects post-flight. Saputra et al. [134] studied the effect of open Space Tourism on regional development, which is measured by tourist satisfaction, destination image, and visitor attraction and commitment. In addition, the public increasingly views Space Tourism not just as a kind of entertainment but also a resource that provides access to alternative airspace, provides new opportunities for scientific and academic research, and creates opportunities for technological development and innovation [135]. Ganesha et al. [136] applied sentiment analysis from Twitter data to analyze the opinion of people about Space Tourism and found that the majority of people have confidence in space travel and eagerly await the day when they can fly to space, whereas few individuals worry and have negative feelings about the potentially harmful impacts of Space Tourism. Security is an importance issue for spaceflight [137]. Maintaining the sustainability dimensions in Space Tourism is another critical matter of concern [138].

### 4.3.10. Cultural and Heritage Tourism

Cultural and Heritage Tourism are among the major tourist attractions. This parameter covers concepts related to Heritage Sites, Historical Sites, UNESCO, Archaeological and Cultural Heritage, Museums, Visitor Experiences, Exhibitions, Art and History, and Augmented Reality. The parameter highlights the need for the protection of historical sites and the enhancement of tourism involving Cultural and Heritage. The dimensions in this parameter include the development of virtual museums based on emerging technologies [139], building virtual reality systems to develop virtual tours in historical tourist attractions [140], using augmented reality smartphone applications to promote historical tourism [141], developing digital storytelling methods based on recommendation systems to improve the experience of tourists who come into contact with artistic and cultural heritage [142], and educating the next generation about the historical places by using video games [143]. For example, for virtual museums, Garlandini et al. [144] illustrated how museums benefited from digital innovative technologies during the pandemic shutdown and attracted people efficiently in a safe and accessible way. Sapio et al. [143] proposed video games as an entertainment tool for historical places in Italy to teach users and the next generation about such places and enable them to interact with the world safety.

### 4.3.11. Music and Art Tourism

This parameter captures various dimensions of tourism involving music and art festivals. The keywords in this parameter include Festivals, Event, Music, Cultural, Visitor, Art, Community, Impact, Identity, Place, Organize, Culture, Economic, Group, Experience, Finding, City, Social, Held, and Satisfaction. A good number of research articles in this

parameter covers the benefits and harms of tourism related to music events. For example, Carneiro et al. [145] studied the importance of festivals for generating economic impacts on the community and the need to define appropriate strategies to increase the positive impact. Almedia et al. [146] investigated the primary promotional channels for a music festival by understanding the primary marketing communication channels and the most efficient techniques to reach various groups of the public as a vital aspect of festival management. Montoro-pans et al. [147] analyzed the web search behavior of prospective music festival attendees, which indicates hidden patterns of behavior among cultural tourists who want to attend music festivals. Tan et al. [148] examined the influence of music festival-specific factors on visitors' levels of satisfaction and how this impacts people's well-being. On other hand, festivals can also have negative effects on host residents and communities. For instance, Moisescu et al. [149] investigated the negative impacts produced via over-tourism during the time of large music events, which include noise pollution, crime, damage to the natural environment, traffic and parking traffic problems, and others.

### 4.3.12. Religious Tourism

Religious Tourism is a type of tourism where people travel to visit a religious site. The Religious Tourism parameter captures studies related to religious tourism and sites. This parameter is represented by keywords including Religious, Pilgrimage, Spiritual, Pilgrim, Site, Religion, Sacred, Temple, Experience, Visitor, Heritage, Place, Motivation, Cultural, Muslim, Islamic, Tourism, Destination, and Satisfaction. Looking at the documents related to this parameter, we found different dimensions of this parameter, including religious tourist motivations [150,151], religious tourist experiences [152], exploring sacred sites [153], technological developments in religious heritage sites [154,155], and the role of Religious Tourism for enhancing social and economic growth within local communities [156].

### 4.3.13. Halal Tourism

This parameter captures dimensions related to Halal Tourism that is geared toward Muslims to provide them with halal food and other facilities that are compliant with or friendly to the Islamic faith. It includes the following keywords: Muslim, Islamic, Halal, Halal Tourism, Religious, Islam, Destination, Compliant, Friendly, Country, Satisfaction, Religiosity, Religious Tourism, Hotel, Attribute, Finding, Service, and Intention. Religious observances play an important role in the hotel selection process for Muslim travelers, and easily accessible Halal food is a preferred travel requirement for this group. Yen et al. [157] discuss the requirements of Muslim visitors for basic hotel and food services, specify the categories in which these features should be placed, and determine which areas require additional resources to attract more Muslim tourists. Yahaya et al. [158] analyzed the criteria related to observing Islamic rules, such as participating in fasts and prayers, eating halal food, self-discipline and abstinence, refraining from drinking alcohol and illegal sexual practices, and separating people by gender in spas and swimming pools. These are essential requirements for Muslim tourists and hotel visitors and may vary based on the individual Muslim practices.

Dabphet et al. [159] examined the criteria of Muslim tourists' selection of travel locations and accordingly looked into the overall Muslim tourists satisfaction. Rahman et al. [160] investigated the possible effects of non-Muslim visitors' perceptions of halal travel destinations and their word-of-mouth for halal travel destinations. Aji et al. [161] studied Muslims' opinions and plans to travel to non-Islamic nations and determined what variables affect Muslims' decision to travel to non-Islamic countries by examining their perceptions of those countries. Sthapit et al. [162] investigated the overall halal food preferences of non-Muslim travelers, the reasons they previously tried halal cuisines, the pleasant and negative feelings they had, and the memorable aspects of their most recent halal food experiences after returning from vacations.

### 4.3.14. Medical and Wellness Tourism

This parameter captures various dimensions related to tourism that is undertaken nationally or internationally primarily for the purpose of improving wellness and obtaining better or relatively inexpensive treatment of illnesses and healthcare services. This parameter is represented by the following keywords: Medical, Patient, Healthcare, Spa, Hospital, Treatment, Wellness, Service, Healthcare, Travel, Cell, Quality, Factor, Country, Eastern, Industry, and Medicine. We discovered various topics linked to this parameter by analyzing various quantitative data and reviewing the articles related to this parameter. For example, Jangra et al. [163] presented an Internet of Things (IoT)-based framework for health monitoring that may be useful for both medical travelers and hotel management in keeping track of the health of personnel and visitors. It examines numerous bodily vital signals and then informs the admin of each person's health situation. Cham et al. [164] examined how advertisements and social media communications affect hospital branding and trust building before the consumption stage, as well as their effects on medical tourists' perceptions and attitudes about service quality, satisfaction, and behavioral intentions. Mahmud et al. [165] investigated how satisfied and devoted international medical tourists are with the level of healthcare services offered by foreign medical facilities. Prajitmutita et al. [166] identified the most efficient ways for healthcare providers to allocate resources to enhance healthcare quality from the viewpoint of medical tourists. Marković et al. [167] focused on measuring customer satisfaction and service quality, particularly when entering the health tourism industry. It offers guidelines for hospital administrators to create plans that would satisfy patients' demands for high-quality care and make them more competitive in the market for Medical Tourism.

### 4.3.15. Volunteer Tourism

Volunteer Tourism involves volunteering to travel and benefit typically disadvantaged host communities to tackle issues such as health, education, the environment, and the economy [168]. This parameter captures various dimensions of Volunteer Tourism including volunteer tourists' participation, experiences, and motivations during their travel. The keywords represented in this parameter include Volunteer, Experience, Organization, Community, Host Community, Participant, Motivation, Group, Project, International, Practice, Interview, Personal, Cultural, Global, and Program. Examples of studies under this parameter include the participation of women volunteers in tourism in rural areas [169], examining the reasons for volunteer tourists' motivation to participate [170], how volunteer tourism organizations could encourage re-participation motivation in a short-term or long-term period [171], creating volunteering opportunities for visitors and local volunteers engaging in community festivities [172], and how volunteering tourists gain positive benefits related to their social selves and careers [173].

### 4.3.16. Temporal Progression (Tourism Types)

Figure 12 shows the temporal evolution of research intensity for the macro-parameter Tourism Types. The plot shows that there is a somewhat sustained increase in the research activity among all parameters (consider the y-axis scale). The parameter Rural Tourism shows the largest increase in research activity followed by Medical and Wellness Tourism. Obviously, COVID-19 has affected the growth of tourism from 2020; however, its relationship with the decline in the number of research articles needs to be established. Furthermore, we note that the decline in the numbers towards the end of the plots is because the data for 2022 are not for the whole year (the data containing research articles were collected on 30 July 2022, which makes up 58.3% of the annual data). Moreover, another interesting trend is that Rural Tourism shows a higher rate of increase than other types of tourism, and the rate is even higher around COVID-19 period, which indicates agreement with the trend because of the lower levels of social distancing requirements in open areas.

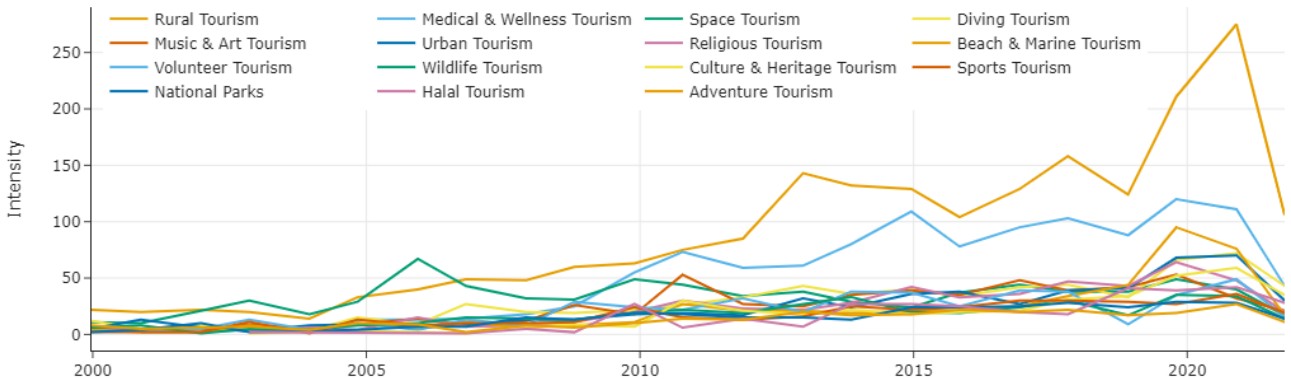

**Figure 12.** Temporal progression (macro-parameter: Tourism types) (data source: Scopus).

*4.4. Tourism Planning*

The macro-parameter Tourism Planning captures the services, planning, and development related to the Tourism Industry. It includes eight parameters, including Education and Training, Workforce, Air Transport, Hotel Industry, Food Services, Economic Growth, Economic Growth for Underdevelopment Communities, and Forecasting Methods.

### 4.4.1. Education and Training

This parameter relates to the development of Educational and Training programs for enhancing the tourism and hospitality industry. The keywords include Student, Learning, Teaching, Hospitality, School, Skill, Educational, Career, Curriculum, Teacher, Academic, Undergraduate, Graduate, Experience, Group, Training, Management, Higher, Industry, and University. The research topics in this parameter include students' perceptions of jobs in the hospitality and tourism industries [174], developing English courses for tourism purposes [175], creating and delivering courses in the field of sustainable tourism development to meet particular training requirements [176], and establishing models of open education for tourist companies by training experts in designing tourism areas and leisure environments [177].

### 4.4.2. Workforce

This parameter covers various dimensions related to the workforce in the hospitality and tourism sector including Workforce required skills, attributes, and behaviors, support structures, satisfaction, management, performance, and productivity. It contains keywords including Employee, Job, Organizational, Hospitality, Work, Hotel, Satisfaction, Performance, Workplace, Leadership, Commitment, Relationship, Working, Training, Service, Skill, Manager, Industry, Effect, and Human Resource. For example, Haarhoff et al. [178] examined the relationship between positive travel experiences and foreign language proficiency of the labor force and discussed issues to better understand how language and tourism interact. They encouraged tourist organizations to pay attention to language challenges and suggested that employment criteria for employees in the tourism sector should also consider language skills. Bani-Melhem et al. [179] indicated the most important factor influencing employees' creative behavior is their workplace satisfaction, with colleagues' support serving as a key mediating factor. Moradi et al. [180] demonstrated how better organizational commitment and e-training will enhance the success of virtual teams working in e-tourism. They also stressed that employees' participation in training initiatives plays a big part in deciding how productive and effective employees are at work.

The staff of international hotels are the most crucial links in the service delivery chain since they interact directly with the hotel's guests. Tsai et al. [181] suggest that international hotel managers must build positive relationships with their internal workforce and be effective future leaders in a dynamic environment. Choi et al. [182] investigated ways to improve the use of smart work by looking at the personal opinions of the employees.

They also demonstrated that employees did not have a comprehensive understanding of smart work effectiveness as one of the environmental protection measures in sustainability management concepts during the COVID-19 pandemic.

### 4.4.3. Air Transport

This parameter captures dimensions related to Air Transport (e.g., operational and management aspects) in connection with the tourism sector. The keywords include Airline, Airport, Air, Low Cost, Aviation, Passenger, Carrier, Cost, Transport, Low, Flight, Service, Travel, Industry, Demand, Route, Market, International, Traffic, and Code. This dimension and research regarding this parameter include the enhancement of airport infrastructure to meet the increased demand of passengers [183], ensuring the accessibility of direct flights to and from a location to improve the number of tourists arriving [184], measuring airport security procedures and their effects on travelers' selection of destinations during the COVID-19 outbreak [185], evaluating international tourist satisfaction when using online reservation services [186], and analyzing how low-cost airlines set their prices and how the Internet affects this strategy where both users and businesses profit from the use of internet in the purchase and sale of airline tickets and looking for the most reasonably priced flights [187].

### 4.4.4. Hotel Industry

This parameter covers various dimensions related to the issues and enhancement of the Hotel Industry in the hospitality and tourism sector. It contains keywords including Hotel, Green, Customer, Hotel Industry, Hospitality, Service, Performance, Guest, Quality, Room, Management, Efficiency, Business, Star, Manager, Environmental, Practice, Result, and Factor. The dimensions captured by this paper include, among others, the use of technology to provide better hospitality services, the use of technology in compliance with safety and social distancing protocols during pandemics, and the use of social media in marketing and engaging tourists, information security, and crisis management.

For example, effectively utilizing advanced and emerging technologies in the Hotel Industry influences travelers' behavior and their positive reactions to these technologies. Çakar and Aykol [188] investigated how robotic services greatly increase the quality of service provided to travelers while also positively influencing travelers' decision to return to robotics-enabled hotels in the context of customer engagement behaviors. Furthermore, in circumstances where social distance is necessary, the employment of robots in hospitality and tourist businesses will boost the mobility of individuals seeking to travel by implementing social separation through the use of robots in services. Chen et al. [189] indicated that hotel managers have to leverage social media networks to attract potential guests, since social media networks have become key contributors to customers' decision-making process when booking and visiting a hotel. Wang et al. [190] clarified how the information security policy attributes are essential for the hotel sector.

Dealing with crisis management is essential in the hotel industry. Murad et al. [191] determined the crisis management strategies employed by five-star hotels and discussed the coping and reaction methods employed to manage these crises. Martinez et al. [192] stress that hotel companies should seek to establish a green positioning strategy that can create goods and services that have both high-value and green qualities.

### 4.4.5. Food Services

This parameter captures research related to Food Services in the hospitality and tourism sector. Food services are an indispensable part of the tourism sector, and many see it as two sides of the same coin [193]. It includes the following keywords: Food, Restaurant, Cuisine, Destination, Gastronomy, Product, Finding, Consumption, Hospitality, Culture, Implication, Safety, Service, Satisfaction, Intention, Industry, Result, Image, and Quality. Examples of topics discovered in this parameter include the investigation of using robots in the food and beverage industry to overcome the lack of workers [194],

evaluating the quality of services and customer satisfaction in the restaurant industry in which physical architecture, restaurant design, cost of the products, and the staff's response are significant factors of customer satisfaction [195], minimizing food surplus and waste in restaurants [196], the requirements of religious visitors for basic hotel and Food Services [157], and proposals for effective tourism-related food safety planning and policy [197].

### 4.4.6. Economic Growth

This parameter highlights the positive contribution of the tourism sector to the economic development of a country. It is represented by the following keywords: Growth, Economic Growth, Economic, Income, Country, Causality, Run, Effect, Demand, GDP, Arrival, Relationship, Model, Result, Long, Test, Long Run, Panel, and Domestic. The parameter captures various dimensions of economic growth including sustainable tourism, visitor satisfaction, host-nation well-being, safety, and prosperity, meeting tourist demands, and more. For example, Kotlyarov et al. [198] emphasized the importance of working toward the goals of sustainable development of the tourism industry to assure a high level of visitor satisfaction and achieve a balance between the host nation, the tourists, and the environment. This can lead to providing new jobs and a comfortable and safe environment, which improves the national economics and industry. Hussein et al. [199] stressed that the development of educational tourism while taking the university quality, pricing decisions, and student demand into account will make the country a center for higher education and foster economic growth and sustainability. Al-hammadi et al. [200] investigated the relationship between Halal Tourism and economic development, which has become an internationally attractive sector as a result of increased demand not only from Muslim visitors but also from non-Muslims. They stated that due to high demand, many Muslim and non-Muslim nations have taken the initiative to innovate and diversify their tourism industries by promoting Halal tourism in order to attain the satisfaction and commitment of travelers.

The number of international tourists plays a critical role in promoting Economic Growth. Din et al. [201] suggested that the economy, the number of global natural and cultural heritage sites, ethnic diversity, and strong governance are major variables affecting international tourists' destination choices. Tourism development must be based on the "green economy", which will improve job opportunities, socioeconomic growth, the protection of natural, cultural, and architectural heritage sites, and the utilization of natural resources while ensuring the renewability and sustainability of consumption [41].

### 4.4.7. Economic Growth for Underdeveloped Communities (EGUC)

This parameter captures tourism dimensions related to the role of tourism in developing and uplifting economically poor communities. The captured dimensions allow governments, policy makers, system designers, and operations professionals to learn the opportunities and challenges in this area and develop better solutions. The parameter includes the following keywords: Poverty, Poor, Community, Income, Economic, Country, Growth, Sector, Reduction, Government, Benefit, Low, Development, Rural, Policy, Household, Level, Social, and Political. Tourism can play a major role in developing low-income communities. Several studies have examined the issues and relationships between tourism and low-income communities. Zeng et al. [202] examined the impact of tourism on the income of poor communities and how tourism development can benefit low-income regions. Sati [203] explained that the construction of new infrastructure including hotels and homestays creates numerous job opportunities for the local community and moves it to above-poverty levels. Cole [204] investigated how tourism assists a poor, distant community's growth, and how the village benefits from the advantages of tourism without experiencing tourism's potential drawbacks. Lu et al. [205] studied the strategies that the government took to encourage rural rejuvenation through tourism, particularly in places

of extreme poverty. Holden et al. [206] investigated how an understanding of poverty in communities helps provide alternative, tourism-related livelihood opportunities.

### 4.4.8. Forecasting Methods

The parameter captures various Forecasting Methods, research, and practices in the tourism sector. It is represented by keywords including Forecasting, Model, Forecast, Demand, Method, Google, Prediction, Neural, Time Series, Neural Network, Time, Series, Algorithm, Network, Performance, Arrival, Accurate, Combination, Result, and Error. Extensive research has been carried out on the forecasting of tourism-related variables. Examples include tourist arrival forecasting [207], rainfall and natural disaster forecasting [208], weather and temperature forecasting [34], forecasting museum visitor behaviors [209], designing tourist infrastructure, project planning for accommodations and transportation development, and reliable tourism demand forecasts to predict the number of international tourists for government agencies [210].

### 4.4.9. Temporal Progression (Tourism Planning)

Figure 13 shows the temporal evolution of research intensity for macro-parameter Tourism Planning. The plots show the overall growth in all parameter activities, with the highest activities in Education and Training followed by Food Services and the Hotel Industry. Note that the decline in the numbers towards the end of the plots is because the data for 2022 are not for the whole year as we collected scientific abstract research for the years of 2000 to 2022. As mentioned earlier, COVID-19 has affected the growth of tourism from 2020; however, its relationship with the decline in the number of research articles needs to be established. Moreover, the decline in the numbers toward the end of the plots is because the data for 2022 are not for the whole year (the data contained in research articles were collected on 30 July 2022, which contributes 58.3% of the annual data).

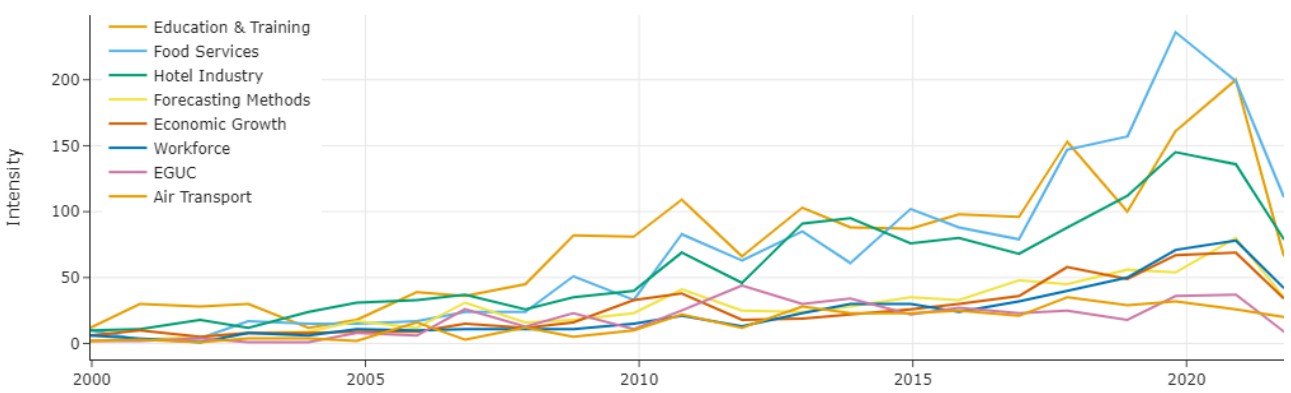

**Figure 13.** Temporal progression (macro-parameter: Tourism planning) (data source: Scopus).

### 4.5. Tourism Challenges

The macro-parameter Tourism Challenges covers major challenges facing the tourism sector. It includes six parameters: Climate Change, Air Quality and Pollution, Water Quality and Pollution Management, Disasters, Pandemics, and Sustainable Tourism.

### 4.5.1. Climate Change

The Climate Change parameter captures the problems and solutions surrounding the effects of the tourism and hospitality sector on Climate Change and vice versa. Climate Change and the current state of tourism have serious consequences for each other. This parameter is created from merging clusters 20 and 44. It consists of the following keywords: Climate, Weather, Winter, Resort, High, Temperature, Mountain, Season, Impact, Adaption, Risk, Destination, and Others. The dimensions and challenges covered by this parameter include extreme weather conditions [35], increasing Climate temperature [32–34], rising

sea levels [211], measurement of the ozone level [212], the reduction of carbon [213], and others. For instance, an example of the extreme weather conditions dimension is the work by Younes et al. [32] who proposed a driver assistance system that recommends a safe speed during extreme weather conditions. Moreover, flight delays depend on weather conditions, and Liu et al. [214] investigated a wide range of variables that might potentially impact flight delays and suggested a gradient boosting decision tree (GBDT)-based model for generalized flight delay prediction. Hernández-Travieso et al. [34] proposed a system with a broad strategy for obtaining an accurate temperature forecast by using artificial neural networks. Gazioğlu et al. [211] examined the relationship between sea level rise (SLR), depression, and the effects of these phenomena on freshwater supplies that could be more vulnerable to the saltwater incursion. Groundwater resources are heavily utilized by the local population for urban, tourist, and agricultural water usage, which poses a serious threat to the coastal aquifer recharge.

### 4.5.2. Air Quality and Pollution

The Air Quality and Pollution parameter is represented by the following keywords: Carbon, Emission, Low, Energy, $CO_2$, Environmental, Consumption, Policy, Country, Travel, Economic, Greenhouse, Gas, and Others. The parameter relates to air pollution resulting from tourism activities where the Air Quality has a significant impact on tourism decision-making before and during the trips. International tourism is a primary source of carbon emissions [215], and traffic and public transport affect environmentally friendly development in the transportation industry [216]. Furthermore, the aircraft emissions during the cruise cycle and the landing/take-off cycle affect the Air Quality [217]. The use of green technology n the transportation sector is expected to minimize air pollution and is an active area of research.

### 4.5.3. Water Quality and Pollution Management (WQPM)

This parameter captures the effects of tourism on water availability, consumption, and pollution, and vice versa. It is represented by keywords including Water, Lake, River, Water Resource, Area, Quality, Resource, Basin, Pollution, Flood, Concentration, Management, Supply, Plant, Environmental, Ecosystem, Flow, and Source. Looking at the material produced by BERT including the documents that belong to this parameter we were able to find various dimensions of this parameter including sustainable water resource management [218], the impact of tourism activities on Water Quality and Water Pollution [219], water consumption in tourist pools [220], water quality prediction systems [221], and overcoming water scarcity via the treatment of wastewater and reuse of water [222].

### 4.5.4. Disasters

This parameter highlights how the tourism sector can be vulnerable to natural disasters and how it can solve the challenges from the Disasters that have already occurred and the risk of future Disasters. The parameter is created by merging clusters 35 and 28. It is represented by keywords including Tsunami, Disaster, Earthquake, Damage, Recovery, Flood, Area, Hazard, Coastal, Affected, Tourism, Vulnerability, Resilience, Building, Impact, Death, War, Site, Heritage, Museum, and Visitor. The dimensions and topics of the papers contained in this parameter include the impact of Disasters on tourism landmarks and the effects of frequent natural Disasters on tourist flow [223], forecasting natural Disasters [208], Disaster management recovery [224], tourist behavior after a recent Disaster in their planned destination, and how a country develops protection against potential natural Disasters [225]. For example, to protect cultural heritage from natural Disasters, Uysal et al. [226] developed a model that generates 3D photo-realistic model using photogrammetry methods for the registry. Kovačić et al. [227] investigated whether tourists' behavior is influenced by perceived risks when deciding to travel to disaster-impacted destinations. Gain et al. [224] examined the planning and recovery actions that the government has to consider during natural Disaster and crisis periods in India.

### 4.5.5. Pandemics

This parameter captures the impact of COVID-19 and other Pandemics on the tourism sector. It is represented by the keywords COVID-19, Pandemic, Crisis, Disease, Industry, Health, Travel, Impact, Outbreak, Risk, Sector, Affected, Business, Measure Global, Countries, and Policy. The COVID-19 pandemic has affected every element of human life, including tourism. The impression of tourism during and after a Pandemic has been altered by policies and travel restrictions established in numerous countries. The dimensions and issues related to this parameter include a sharp reduction in the number of tourist arrivals [228–230], increased travel costs and effects on the restaurant industry [231], airlines industry [232], lockdown of places of religious tourism [233], and almost entire national economies [234,235].

For example, Wang et al. [230] developed effective techniques to help predict tourist arrivals after the COVID-19 Pandemic. Ghosh et al. [229] studied how international tourism in Australia was influenced by the pandemic. They investigated the primary factors of tourism in favorable times and the principles that may be derived to re-establish international travel in the post-pandemic world. Polese et al. [231] demonstrate the importance of using technology to establish a service ecosystem supporting restaurant management. Moreover, the government set new rules and policies to overcome the COVID-19 Pandemic.

### 4.5.6. Sustainable Tourism

Sustainable Tourism is defined by the WTO (World Tourism Organization) as "tourism that takes full account of its current and future economic, social and environmental impacts, addressing the needs of visitors, the industry, the environment and host communities" [7]. This parameter captures various dimensions of Sustainable Tourism including Environment, Sustainability, Ecotourism, Communities, Destinations, Visitors, Residents, Homes, Impacts, Management, People's Behaviors, Norms, and Attitudes. The research activities in this parameter include, among others, managing ecotourism by protecting natural resources and environments, preserving traditional cultures of the host communities, and enhancing the environmental protection awareness of the local population [5,6], tourists' attitudes toward sustainable consumption and purchasing habits concerning eco-friendly products [8], managing over-tourism and the impacts on residents [9], and citizens' participation in environmentally sustainable actions [10]. For example, Schubert et al. [9] asserted that over-tourism has a negative impact on the cultures of the residents and consumption such as booked-out restaurants and crowded hiking trails. They stressed that maintaining investments in high-quality tourism while minimizing numbers will increase locals' well-being.

### 4.5.7. Temporal Progression (Tourism Challenges)

The temporal evolution of research intensity for the macro-parameter Tourism Challenges is shown in Figure 14. The plots show that there is a somewhat sustained but relatively low increase in research activity among all parameters, except Pandemics and Water Quality and Pollution Management (WQPM) (consider the y-axis scale). The relatively low growth could be attributed to the 2007 recession. The sharp increase in the Pandemics parameter is expected due to COVID-19. The relatively higher growth in the research intensity for Water Quality and Pollution Management (WQPM) reflects the significance of the problem dimension. The decline in the intensity toward the right side of the plots was explained earlier.

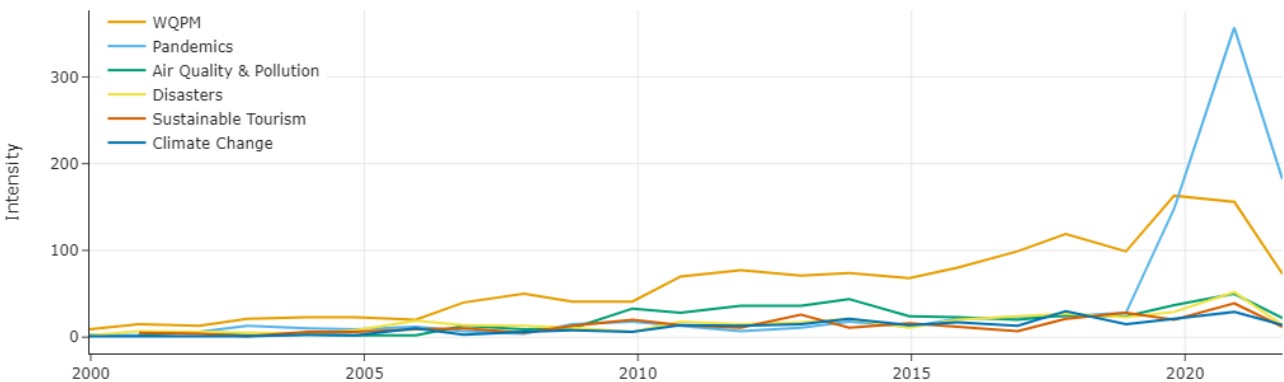

**Figure 14.** Temporal progression (tourism challenges) (data source: Scopus).

### 4.6. Media and Technologies

The Media and Technologies macro-parameter relates to various uses of Media and Technologies in development, marketing, and other functions of tourism. The use of media and technologies contributes to enhancing services and service quality, improved operational effectiveness, enhanced customer satisfaction, and cost savings. It includes the following parameters: Film and TV Media, Online and Social Media, Mobile Applications, and Virtual Reality.

#### 4.6.1. Film and TV Media

This parameter captures the role of Film and Television Media in exploring and marketing popular tourist locations and destinations. The keywords related to this parameter include Film, Television, Induced, Destination, Image, Location, Media, Popular, Series, Viewer, Screen, Audience, Culture, Place, Phenomenon, Destination Image, Visit, and Involvement. Bolderman et al. [236] investigated the strategies that popular media produces to generate new tourism flows and emphasized how movies and music could engage the geographical imagination and physically move tourists to these locations. Nieto-Ferrando et al. [237] examined the impact of Movie-Induced Tourism (MIT) stereotypes on the satisfaction of visitors who visited a place after watching films. Iwashita et al. [238] explored the importance that films and television dramas play in influencing international visitors in selecting their travel destinations. Garrison et al. [239] studied the relationship between media tourism and sustainability and their influence on rural communities. Kay [240] used music as an instrument to promote tourism.

#### 4.6.2. Online and Social Media

The Online and Social Media parameter captures the significant impact of Social Media applications in the tourism sector and the way it has transformed traveling and tourism. The keywords represented in this parameter include Online, Travel, Website, Social, Media, Satisfaction, Review, and Others. The tourism industry is becoming more influenced by the spread of Social Media and user-generated content. Consumers share their tourism and hospitality experiences with other consumers online by posting reviews of their stays. The articles related to this parameter disclose various dimensions of using Social Media in tourism. These dimensions include automatic text analysis and mining [76,241,242], forecasting tourist demand [24,243,244], tourist recommendation systems [245], and mining travel locations and routes [246].

#### 4.6.3. Mobile Applications

Mobile Applications have a significant impact on tourism activities. The Mobile Applications parameter captures various developments and services that make use of mobile devices and applications. It is represented by keywords Including Mobile, Applications, Smartphone, Location, Services, Technology, and Maps. Looking at the scientific papers

that belong to this parameter, we were able to find several topics that capture various mobile application services. These services include integration of the Android platform and GPS services in guiding tourists to their preferred sites [29], augmented reality for historical tourism using mobile and smart devices [30,31], the use of Mobile Applications for tracking tourist travel experiences [247], determining different types of tourists mobility and providing location-based services [20,248], and implementing QR systems for shopping centers [249].

The use of Mobile devices and Applications together with augmented reality has transformed visitor experiences by providing a chance for meaningful connection with distinctive cultures and history on smartphones. Jiang et al. [31] investigated the effectiveness of augmented reality (AR) in promoting the memorability of tourist experiences at heritage sites such as the Great Wall of China, utilizing a smartphone app. Location-based services are used as part of information systems for tour guides to provide travelers with route planning and tourism information. Yang et al. [248] developed a master multi-agent system utilizing picture recognition technology and Google Maps as a source of tourism information and a route planner for travelers. It combines a smartphone GPS function, a QR/Bar code scanner, and access to a cloud database, allowing users to locate all the necessary web services while traveling for business or pleasure. Huettermann et al. [250] discuss how visitors may use smartphone apps and obtain real-time forecasting information to resolve issues such as parking, traffic, and queues.

### 4.6.4. Virtual Reality

The Virtual Reality parameter is created by merging two clusters 7 and 34. The parameter is depicted by the keywords Virtual, Reality, 3D, Video, Camera, Technology, Augmented, Image, Site, Cultural, Urban, and Data. The parameter captures several dimensions represented by Virtual Reality applications in the tourism sector. These applications include web-based visualization and 3D modeling for cultural heritage objects [251], displaying a 3D image at tourist destinations [252], a virtual city tour [253], a cultural entertainment system that allows tourist to improve their travel experience by carrying out a series of games based on augmented reality [254], and virtual reality applications for museum visitors. For instance, applications were reported in [255] to visualize the grand mosque in Medina (Al-Masjid Al-Nabawi) and religious places in an interactive style with ease of use for guests with the interaction of 3D environments.

Automatic text analysis focuses on mining tourist information to improve the quality of experience for both tourists and the public tourism industry. For example, García-Pablos et al. [241] applied sentiment analysis of hotels to measure tourist satisfaction and feelings related to other factors. Al Sari et al. [76] investigated the role of cruises in Saudi Arabia using social media platforms and a machine learning algorithm. Feizollah et al. [256] applied sentiment analysis based on Twitter data, which analyzed tweets related to Halal tourism.

As regards forecasting tourist demand, for instance, Colladon et al. [24] applied social networks and predicted foreign arrivals at airports in the capital cities of Europe. Yuan et al. [257] predicted popular tourism locations and travel routes to help users obtain a better travel schedule.

This parameter also captures the impact of the COVID-19 pandemic on the hotel and lodging sector by providing a comparison of guest satisfaction before and after the pandemic [258]. The impact of COVID-19 on the airline industry was reported in [259].

### 4.6.5. Temporal Progression (Media and Technologies)

Figure 15 depicts the temporal progression of the macro-parameter Media and Technologies. The vertical line of the graph indicates the number of research articles, defined as the intensity. We note that the activity in the Mobile Applications parameter has significantly increased over time compared to other parameters followed by the Online and Social Media and Virtual Reality parameters. The Film and TV Media parameter has seen

some growth after around 2010, although the growth is small compared to the other three parameters. These trends are in line with the expected trends and potentials of these media and technologies. Virtual reality is a high-potential technology, and its relatively low intensity is due to virtual and augmented reality applications covered under mobile applications research. The discussions about the parameters included in this macro-parameter and the plots in the figure clearly show the evolving nature of societies in relation to tourism; there is an increasing tendency of society to use mobile smart devices, social media, and virtual reality for tourism with relatively low usage of film and TV media.

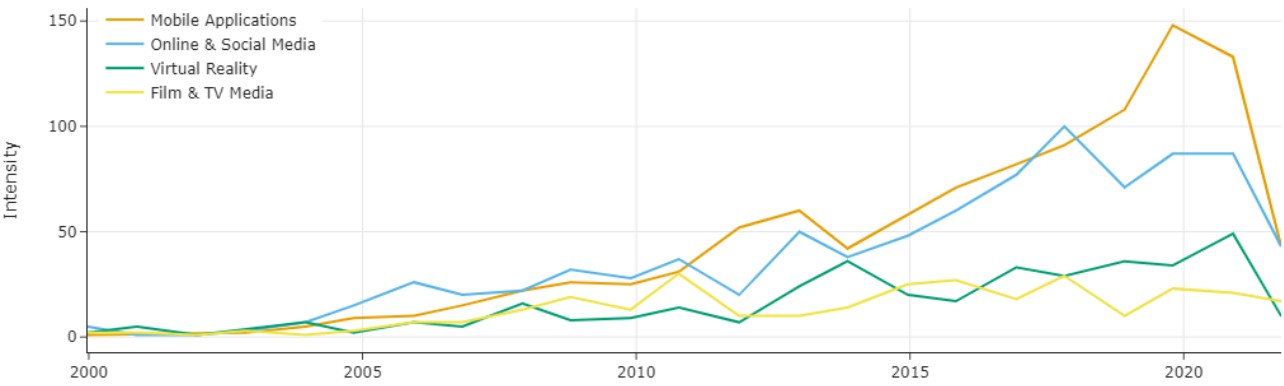

**Figure 15.** Temporal progression (Media and Technologies) (data Source: Scopus).

Note that some of these trends (see the lines 2019/2020 onwards) are influenced by COVID-19. The sharp drop at the end is due to the data period that does not include scientific articles for the entire year of 2022.

## 5. Tourism Parameter Discovery (Public: Twitter)

This section discusses the tourism discovered parameters by our BERT model from the Arabic Twitter dataset. It captures the national perspective of tourism about Saudi Arabia. The parameters are grouped into two macro parameters: Tourist Attractions and Tourism Services. We provide an overview of the parameters and taxonomy in Section 5.1. The quantitative analysis is discussed in Section 5.2. Subsequently, we discuss each macro-parameter in separate sections, namely Sections 5.3 and 5.4.

### 5.1. Overview and Taxonomy

We discovered a total of 30 clusters using Bert Topic modeling. Then we analyzed the results and excluded irrelevant clusters that did not have important information for our analysis. We removed six clusters due to their irrelevant themes. The remaining clusters merged as necessary and resulted in a total of 13 parameters. The parameters were grouped into two macro-parameters based on our domain knowledge, similarity matrix, hierarchical clustering, and other quantitative methods. In this section, we discuss the methodology and process used to discover parameters and group them into macro-parameters.

Table 3 lists the parameters and the macro-parameters discovered by our BERT model from the Arabic Twitter dataset. The parameters are grouped into two macro-parameters (Column 1), namely, Tourist Attractions and Tourism Services. The second and third columns indicate the parameters and their number, respectively. The fourth column lists the percentage of the number of documents. Our BERT model classified 42.7% of tweet documents in outlier clusters and we ignored these clusters, and the remaining 49.5% of tweet documents are listed in the fourth column. The fifth column lists the top 10 keywords associated with each parameter along with their English translations. As part of our efforts to gain a better understanding of the parameters, we examined the tweets associated with each parameter. As shown in the following table, we also contextually translated the Arabic tweets' content so that English readers can better understand the content.

Figure 16 shows a taxonomy of the tourism parameters that were discovered by our system. The taxonomy was created from Table 3, and it indicates the tourism parameters, their macro-parameters, and some keywords related to the parameters. The first-level branches represent the macro-parameters including Tourist Attractions and Tourism Services. The second-level branches represent the discovered parameters such as International Destinations, National Destinations (Saudi Arabia), NEOM, Programs, Tours and Packages, Restaurants and Hotels, etc. The third-level branches represent the most representative keywords associated with each parameter.

**Table 3.** Macro-parameters and parameters for tourism (data source: Twitter).

| Macro | Parameters | No | % | Keywords |
|-------|-----------|----|----|----------|
| Tourist Attractions | International Destinations | 1 | 3.04% | أذربيجان، قابلا، قوبا، باكو، بورجومي، دبي، باتومي، تبليسي، قطر، البحرين<br>Azerbaijan, Qabil, Quba, Baku, Borjomi, Dubai, Batumi, Tbilisi, Qatar, Bahrain |
| | National Destinations (Saudi Arabia) | 0 | 14.20% | السياحة، السفر، الرياض،نيوم، الداخليه، موسم ، العالم ، جده المدينه ، العلا<br>Tourism, Travel, Riyadh, Neom, Local, Season, The World, Jeddah, Medina, Alula |
| | | 25 | 0.85% | خيبر، بدر، ينبع، العلا، المنوره، المدينه، جازان، البكيرية، للتفاصيل، استعلم<br>Khyber, Badr, Yanbu, Alula, Munawwarah, Medina, Jazan, Bukayriah, Details, Inquire |
| | NEOM (Smart City) | 8 | 1.03% | نيوم، الأكثر، الاحلام، مستوي، المشروع، طموحا، العالم ، للقائد، والملهم، المستقبل<br>NEOM, The Most, Dreams, Level, Project, Ambitious, World, Leader, Inspiring, Future |
| | | 13 | 0.92% | نيوم، جده، مطار ، الرياض ، برج ، المراقبه ، فرضت، مغلقه، واشنطن، المرضي<br>NEOM, Jeddah, Airport, Riyadh, Tower, Observation, Imposed, Closed, Washington, Accepted |
| | | 22 | 0.66% | المستقبل، والاستثمار، نيوم، يعلن، تصاميم، اهداف، أطلقها، لاين، الاقتصاديه، فكرتها<br>Future, Investment, NEOM, Announces, Designs, Goals, Launched, Line, Economic, Idea |
| | AlUla City | 3 | 2.14% | العلا، املج ، قمة ، مدائن ، جده ، قرارات، اعلى ، الخليجي ، وزراء، بتبوك<br>AlUla, Umluj, Summit, Mada'in, Jeddah, Decisions, Higher, Gulf, Ministers, Tabuk |
| | Abha City | 14 | 0.96% | ابها، عاصمه، طرق ، لعام ، ولمنطقه ، للمدينة ، انجاز، يعتبر ، عسير ، ينبع ، السياحه<br>Abha, Capital, Roads, Year, Region, For City, Achievement, Considered, Asir, Yanbu, Tourism |
| | Jeddah Historical | 18 | 0.75% | التاريخيه، جوله، جده، تستضيف، والجغرافية، بمكانتها، أهميتها، مساعده، مساعد، الجيولوجية<br>Historical, Tour, Jeddah, Host, Geographic, Position, Importance, Assistance, Helper, Geologic |
| | Natural Places | 26 | 0.55% | القوقاز، جبال ، الأوربية ، الطبيعه ، جنه ، المسافر ، يلمسها، امتزاج، حلما، الخلابة، تمتع<br>Caucasus, Mountains, European, Nature, Paradise, Traveler, Touches, Mix, Dream Wonderful |
| | Seasonal Festivals | 11 | 0.99% | تذاكر، احجز، ونترلاند، موسم ، جده ، الرياض ، البوليفارد، يحي، تتوفر ، الي يها، والبليفارد<br>Tickets, Book, Winterland, Season, Jeddah, Riyadh, Boulevard, Come, Available, If You Want |
| | | 16 | 0.92% | تذكرتك، فعاليات، احجز ، جنغل ، جده ، تذاكر ، حفل ، حتعيش، تتاخر ، والموسيقى<br>Your Ticket, Events, Book, Jungle, Jeddah, Tickets, Party, Live, Be Late, Music |
| | | 17 | 0.91% | السنه ، تذاكر، راس، موسم ، الرياض ، حفله ، البوليفارد، العلم ، جزيره ، ونترلاند<br>Year, Tickets, Head, Season, Riyadh, Party, Boulevard, Flag, Island, Winterland |

**Table 3.** *Cont.*

| Macro | Parameters | No | % | Keywords |
|---|---|---|---|---|
| Tourism Services | Programs, Tours & Packages | 5 | 1.49% | سياحي، سياحيه ، برامج ، سيارة ، برنامج ، انسب ، حجز فنادق ، تأجير، متميزة ، جولات ، حجوزات<br>Tourist (a person), Tourist (adjective), Programs, Car, Program, Most Suitable, Hotel Reservation, Rental, Distinguished, Tours, Reservations |
| | | 15 | 0.93% | كشتات، مرافقه ، بوكنيك ، توصيل ، حجز، سائق، شقق ، دليل ، المطار، بأسعار<br>Camping, Accompany, Booking, Delivery, Reservation, Driver, Apartments, Guide, Airport, Prices |
| | | 27 | 0.55% | المسافرون ، رحلات ، العرب ، عمان ، عوائل ، باتومي ، عطلات ، البحرين ، للحجز، قطر<br>Travelers, Trips, Arabs, Oman, Families, Batumi, Holidays, Bahrain, For Reservations, Qatar |
| | Offers, Discounts & Gifts | 29 | 0.50% | عرض ، سافر ، احجز ، واستمتع ، ووفر ، بأسعار ، سفر ، قطر ، منتجع ، جميل<br>Offer, Travel, Book, Enjoy, Save, Prices, Travel, Qatar, Resort, Beauty |
| | | 2 | 2.22% | أذربيجان ، خاصه ، خصومات ، السياحه ، للشركات ، رياده ، وهدايا ، الكبيرة ، استفسار ، الشتاء<br>Azerbaijan, Special, Discounts, Tourism, Companies, Entrepreneurship, Gifts, Big, Inquiries, Winter |
| | | 21 | 0.70% | أفضل ، العروض ، الأسعار ، الالكتروني ، والاسعار ، دوله ، عشاق ، اثناء ، أذربيجان ، الخصومات<br>Best, Offers, Prices, Electronic, Prices, Country, Lovers, During, Azerbaijan, Discounts |
| | Tourist Guides | 4 | 1.53% | والاستعلام ، مرشد ، رحلتك ، عروض ، للحجز، الامارات ، البحرين ، عمان ، تبليسي ، قطر<br>Inquire, Guide, Your Trip, Offers, For Reservations, Emirates, Bahrain, Oman, Tbilisi, Qatar |
| | Restaurants and Hotels | 23 | 0.62% | الأماكن، السياحيه ، جهزت ، والمواقع ، عروس ، والمطاعم، كامل ، الأوسط ، مدن ، الخليج<br>Places, Tourist, Prepared, Sites, Bride, Restaurants, Full, Middle, Cities, Gulf |
| | | 24 | 0.58% | أفخم ، الفنادق ، المطاعم ، اسكن ، السائحين ، القريب ، ارقى ، عرض ، دوله ، اجازتكم<br>Most Luxurious, Hotels, Restaurants, Live, Tourists, Nearest, Finest, Offer, Country, Vacation |
| | | 7 | 1.03% | عروض ، خدمات ، أسعار ، الشتاء ، السياحه ، منتجعات، أفضل ، والأمان ، الراحه ، الأماكن<br>Offers, Services, Prices, Winter, Tourism, Resorts, Best, Safety, Comfort, Places |
| | Medical Insurance & Internet Services (MI2S) | 12 | 0.99% | تامين ، واحصل ، رحله ، وشرائح ، انترنت ، اخبروه ، لسائق ، رأيتم ، تاكسي ، نشرت<br>Insurance, Get, Trip, SIM Cards, Internet, Tell Him, Driver, You Saw, Taxi |

### 5.2. Quantitative Analysis (Twitter)

This section presents the quantitative analysis including the keywords score, the Intertopic distance map, hierarchical clustering, and the similarity matrix. A set of keywords are used to represent each parameter; however, not all of them accurately define it. Figure 17 presents the top 10 keywords for each parameter (see Section 3.5). The importance score, or c-TF-IDF, is used to order the keywords. There are 13 subfigures, and in each subfigure, the horizontal line shows the importance score, and the vertical line shows the parameter keywords.

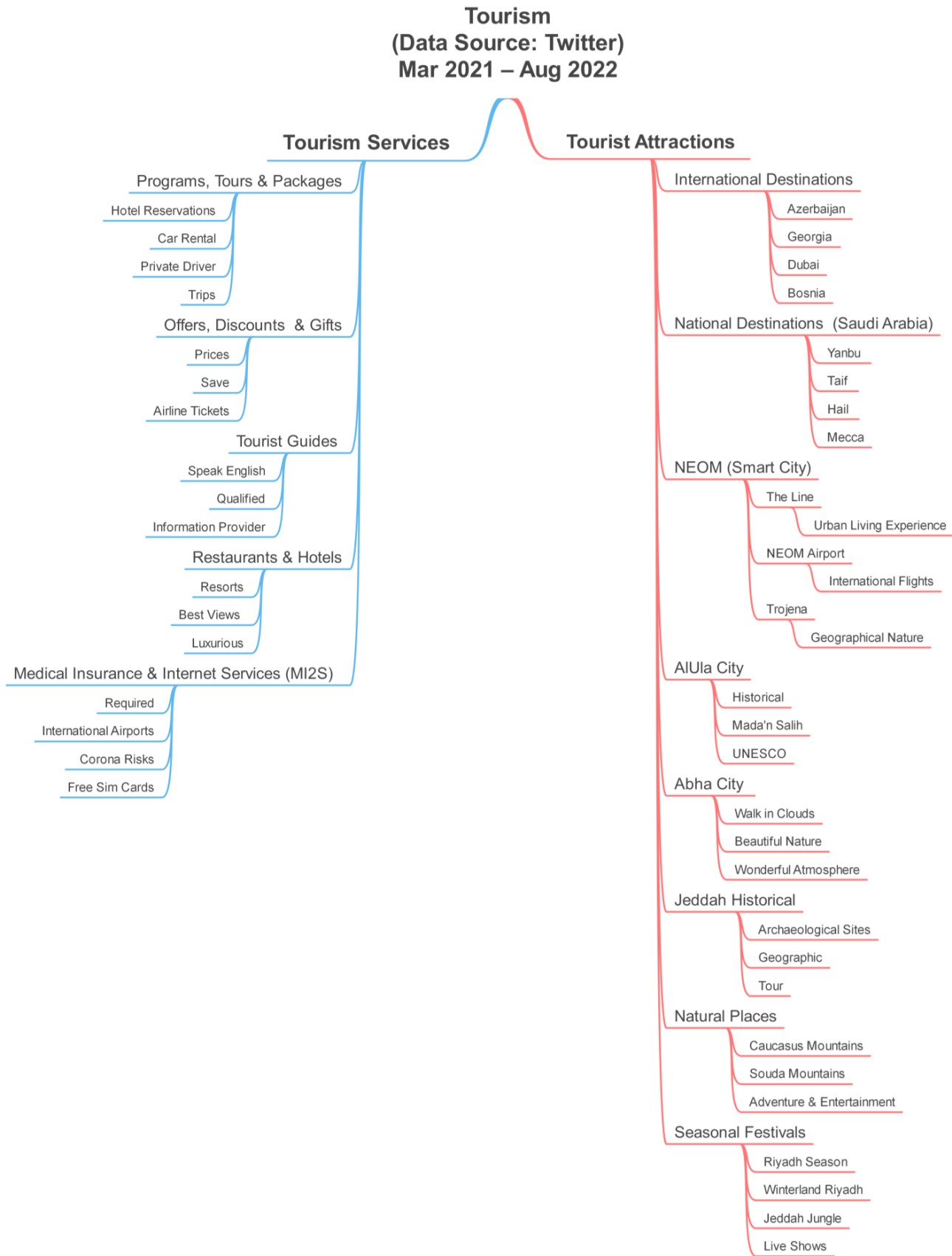

**Figure 16.** Taxonomy of discovered tourism parameters extracted from Twitter data.

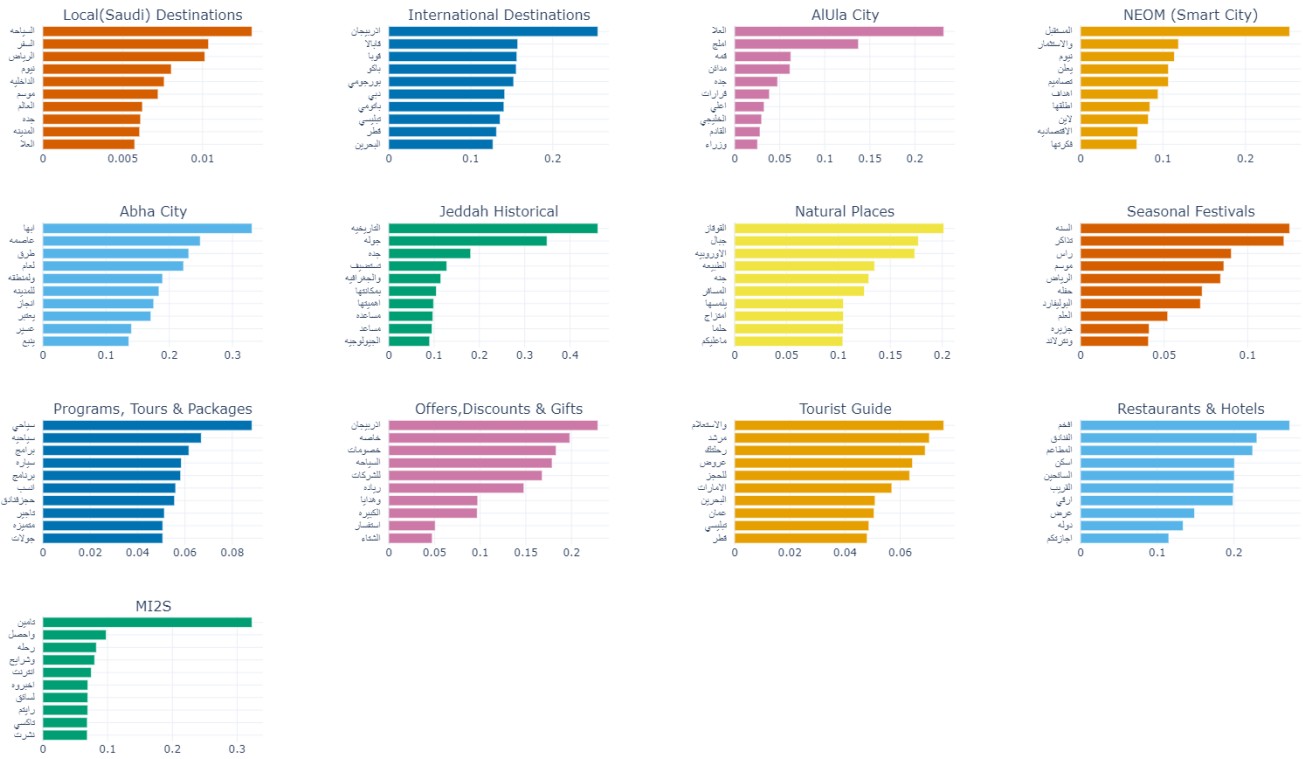

**Figure 17.** Twitter parameters with keywords' c-TF-IDF score (data source: Twitter).

Figure 18 indicates the hierarchical clustering of the 13 parameters and systematically pairs them based on the cosine similarity matrix (see Section 3.5). We noticed that clusters 0, 3, 14, and 22 created a unique cluster that we labelled Tourist Attractions.

## Hierarchical Clustering

**Figure 18.** Hierarchical clustering (data source: Twitter).

Figure 19 shows the similarity matrix among the parameters (see Section 3.5). Dark blue shows the highest similarity between parameters, whereas light green color shows the least similarity. For example, we see a dark blue color between clusters 11 and 18, which indicates a high similarity score because both clusters 18 (Jeddah Historical) and 11 (Seasonal Festivals) are related; Jeddah city had seasonal festivals (Jeddah Season) during the period of 2 May 2022 to 30 June 2022.

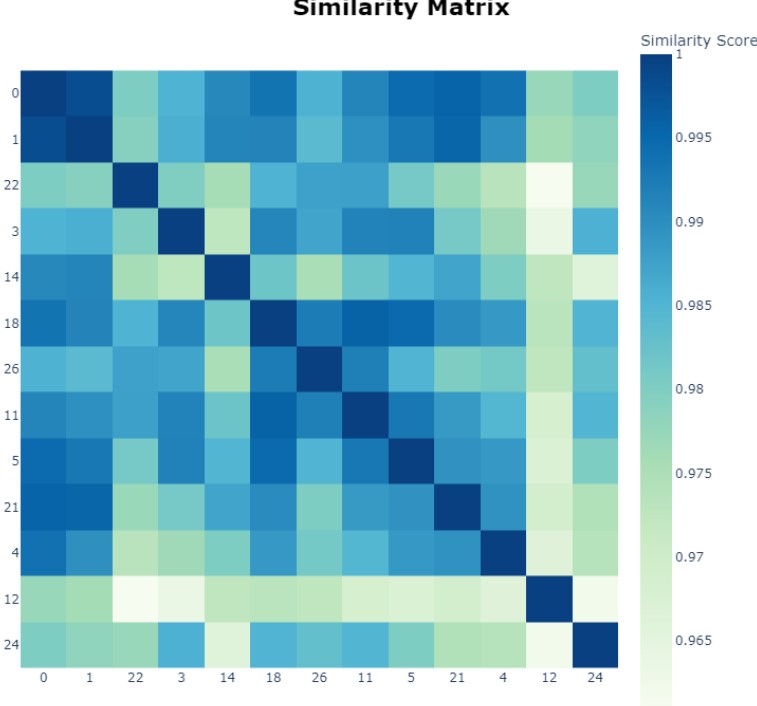

**Figure 19.** Similarity matrix (data source: Twitter).

Figure 20 shows the term scores that identify the number of keywords needed to describe the parameters. It indicated that ten to thirteen terms in each topic's ranking accurately reflect the topic. Since the probability of all other words are so close to one another, their ordering is essentially meaningless.

## Term score decline per Topic

**Figure 20.** Term rank (data source: Twitter).

Figure 21 shows the Intertopic distance map based on a multidimensional scale. The figure clearly identified three macro-clusters.

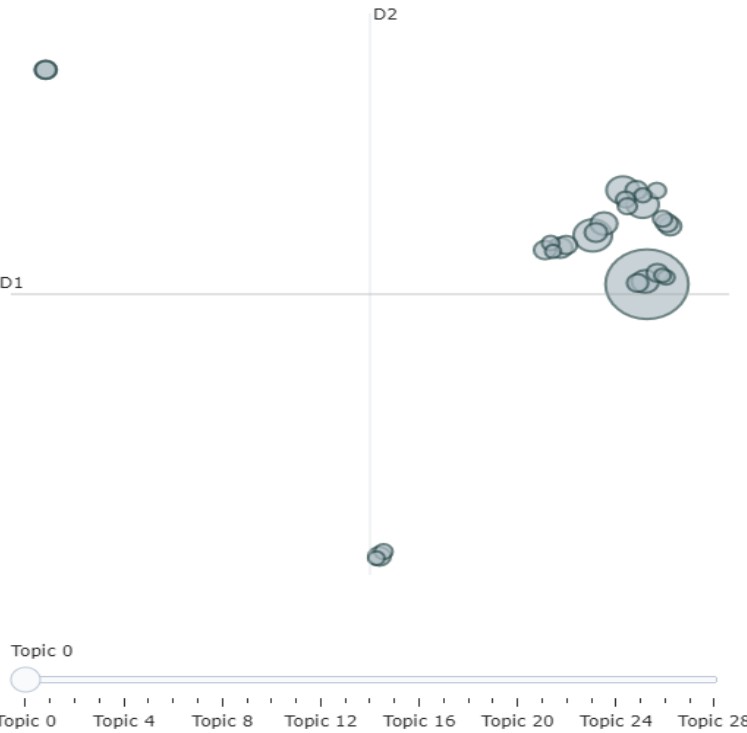

**Figure 21.** Intertopic distance map (data source: Twitter).

*5.3. Tourist Attractions*

In this section, we discuss the first macro-parameter Tourist Attractions. It includes eight parameters, namely International Destinations, National Destinations (Saudi Arabia), NEOM (Smart City), AlUla City, Abha City, Jeddah Historical, Natural Places, and Seasonal Festivals.

### 5.3.1. International Destinations

The first parameter is International Destinations. It is represented by keywords including Azerbaijan, Qabil, Quba, Baku, Borjomi, Dubai, Batumi, Tbilisi, Qatar, Bahrain, travel, Emirates, Tourism, Bosnia, and Trabzon. An example of a tweet related to this parameter from a company is as follows:

"للمزيد من المعلومات عن السياحة في#اذربيجان#باكو#جورجيا#تبليسي#باتومي#قوبا#قابالا#بورجومي#سفر
#سياحة#دبي#الامارات"

*"For more information about tourism in #Azerbaijan #Baku #Georgia #Tbilisi #Batumi #Quba #Qabala #Borjomi #Travel #Tourism #Dubai #UAE"*

### 5.3.2. National Destinations (Saudi Arabia)

The second parameter is National Destinations (Saudi Arabia). It is created by merging two clusters (numbers 0 and 25). This parameter is related to popular tourism destinations in Saudi Arabia. It includes the following keywords: Tourism, Travel, Riyadh, Neom, Local, Season, Jeddah, Medina, AlUla, Mecca, Ministry, Yanbu, Tabuk, and Dammam. The following tweet mentions the hashtags of several tourism destinations in Saudi Arabia. The second tweet is from the Saudi Ministry of Tourism and it highlights the trends towards national and ecological tourism due to COVID-19 risks and physical distancing restrictions:

"#طيران_اديل#السعودية#الرياض#جدة#مكة_المكرمة#المدينة_المنورة#بها#تبوك#حفرالباطن#الدمام#الخبر
#القصيم#بريدة#الزلفي#الجوف#الجمعة#حائل#الطائف#ينبع#نجران#الخرج#الخفجي#العلا#سياحة"

*"Flyadeal #Saudi Arabia #Riyadh #Jeddah #Mecca #Abha #Tabuk #Hafar Al-Batin #Dammam #Khobar #Qassim #Buraydah #Zulfi #Jouf #Al Majmaah #Hail #Taif #Yanbu #Najran #Kharj #Khafji #Ula #Tourism"*

"من الأمور الايجابية التي نتجت عن كورونا هي السياحة البيئية والدينية الداخلية التي نشطت بشكل كبير وهنا دور وزارة السياحة بالتنسيق مع النقابات ووكالات السفر والسياحة"

*"Among the positive things that resulted from Corona is the internal and ecological tourism, which has been very active, and here is the role of the Ministry of Tourism in coordination with unions and travel and tourism agencies"*

### 5.3.3. NEOM (Smart City)

The third parameter is Neom (Smart City), a trillion-dollar smart city being built in the northwest of Saudi Arabia. NEOM intends to bring about a shift in Saudi economy by developing a knowledge-based economy, diversifying the country's economic sources of revenue, and reducing the country's reliance on oil. This parameter is represented by the following keywords: NEOM, Dreams, Project, World, Inspiring, Future, Renewed, Ambitious, Inspired, and Leader:

"مشروع نيوم هو المستقبل والتاريخ الجديد وهو الوجهة الأكثر ملاءمة للعيش على مستوى العالم حيث يسعى لتنمية الاقتصاد السعودي وتنويع مصادره، وتقديم نموذج مثالي لمعيشة مستدامة ومزدهرة."

*"NEOM is the new future and history, the world's most livable destination. It seeks to develop the Saudi economy, diversify its sources, and provide an ideal model for sustainable and prosperous living".*

"رئيس قطاع السياحة في شركة #نيوم من المخطط افتتاح أول فنادق مشروع "نيوم" في نهاية عام 2022، كما سنفتتح ما يصل إلى 15 فندقًا سنويًا في الفترة ما بين 2023 و 2025"

*"Head of the tourism sector in #NEOM: It is planned to open the first hotel in the NEOM project at the end of 2022, and open up to 15 hotels annually between 2023 and 2025"*

"نيوم تتميز بطبيعة جغرافية فريدة من نوعها بالعالم كله ومن اشكال الاستغلال الأمثل لهذي الطبيعه جات مشاريع عملاقة في نيوم مثل اطلاق مشروع #تروجينا واللي راح يصير الاميز في السياحة البيئية"

*"NEOM is characterized by a unique geographical nature in the whole world, and among the forms of optimal exploitation of this nature, giant projects have come together in NEOM, such as the Trogena project, which will become a distinguished name in eco-tourism"*

NEOM will be built around the concept of The Line, a mega project located in NEOM smart city, launched in January 2021. The Line will be a city with a million residents and a length of 170 km that preserves 95% of nature with zero cars, zero streets, and zero carbon emissions [260]. The Line concept has attracted much debate and interest due to its radically different approach towards urban living. It was detected as a separate cluster by our BERT model, but we merged it into one parameter called NEOM. The following are tweets related to this parameter:

"ذا لاين مدينة المستقبل في نيوم، المدينة تستهدف تحقيق مثالية العيش ومعالجة التحديات الملحة التي تواجه البشرية."

*"The Line, the city of the future in NEOM, the city aims to achieve the ideal of living and address the urgent challenges facing humanity"*

"ذا لاين احدى مشاريع المستقبل واللي يخدم البيئة ويرفع من جودة الحياة ويرفع من مستوى السياحة والاقتصاد المحلي"

*"The Line is one of the future projects that serves the environment, raises the quality of life and raises the level of tourism and the local economy"*

Significant Twitter activity was detected around the topic of the Neom International Airport. Our BERT model detected it as a separate cluster. We merged it with the NEOM for knowledge structure and simplicity. Below is an example tweet on the subject.

"انطلقت أولى الرحلات الدولية من مطار نيوم إلى دبي ، خطوة مميزة لتحقيق رؤية المملكة ... شكرا للهيئة العامة للطيران المدني وكافة الجهات"

*"The first international flight departed from NEOM Airport to Dubai, an important step to achieve the Kingdom's vision... thanks to the General Authority of Civil Aviation and all"*

### 5.3.4. AlUla City

The fourth parameter is AlUla City. The Saudi city of AlUla has become one of the new tourist destinations due to its beautiful natural components, diverse history, and antiquities dating back thousands of years. It includes the keywords AlUla, Umluj, Summit, Mada'in, Jeddah, Decisions, Higher, Gulf, Ministers, and Tabuk. The following tweets were posted:

"العلا أعظم مدينه تاريخيةفي المملكة العربية السعودية، وتوجد بها مدائن صالح و هي وجهة سياحية رائعةتتمتع بالعديدمن الثقافات والمواقعالأثرية"

*"AlUla is the greatest historical city in the Kingdom of Saudi Arabia, and there is Mada'in Saleh, which is a wonderful tourist destination with many cultures and archaeological sites"*

"مدائن صالح في #العلا أول موقع سعودي مُدرج على لائحة اليونسكو للتراث العالمي"

*"Mada'in_Saleh in AlUla is the first Saudi site to be inscribed on the UNESCO World Heritage List".*

"محافظة العلا هي تحفه فنيه رائع توجد فيها (مدائن صالح) احد الاثار قديمة والجميلة"

*"AlUla Governorate is a wonderful masterpiece in which there is (Madain Saleh), one of the ancient and beautiful monuments"*

### 5.3.5. Abha City

The fifth parameter is Abha City (also called the Abha Tourist City by Saudi people). The keywords of this parameter include Abha, Capital, Region, Achievement, Asir, and Tourism. The following are examples of tweets related to this topic:

"ابها البهية حباها الله بطبيعة خلابة وصيف ممتع تكتسي فيه سماءها بغيوم متراكمه تشاهدها وانت بأعالي منتزهاتها..."

*"Abha Al-Bahiya. God has blessed it with a beautiful nature and an enjoyable summer in which its sky is covered with accumulated clouds. You watch it while you are in parks above the clouds"*

"مدينة أبها هي عاصمة السياحة العربية لعام 2017م، و الذي يعتبر انجاز للمدينة ولمنطقة عسير بشكل عام"

*"Abha is the capital of Arab tourism for the year 2017. Which is considered an achievement for the city and the Asir region in general"*

"السياحة السعودية: جهود رائعة في صيف أبها لهذا العام من قبل هيّة تطوير عسير نشاطات وفعاليات ومهرجانات متعدد ة استمتع زوار أبها بأجواء رائعة"

*"Saudi Tourism: Great efforts in Abha summer this year by the Asir Development Authority Multiple activities, events, and festivals. Visitors to Abha enjoyed a wonderful atmosphere"*

### 5.3.6. Jeddah Historical

The sixth parameter is Jeddah Historical, also called AlBalad, which means town. It is a historical district of Jeddah city in Saudi Arabia. This parameter is characterized by the keywords such as Historical, Tour, Jeddah, Host, Geographic, Its Position, Importance, Assistance, Helper, Geologic, and Land. The following are some tweets related to this parameter:

"زرت اليوم واستمعت برحلة في تاريخ جده القديمةو مواقعها الاثرية ، وجدت عدداً من السياح وعملت مترجماً لهم لماكتب على اللوحات"

*"I visited today and listened to a journey in the history of ancient Jeddah and its archaeological sites, and I found a number of tourists and worked as a translator for them of what was written on the paintings"*

"جده التاريخيه تعتبر من احد اهم الأماكن التاريخيه في دولتنا واهتمام ولي عهد في تطويرها يعكس حجمه"

*"Historic Jeddah is considered one of the most important historical places in our country, and the interest of the Crown Prince in developing it reflects its importance"*

### 5.3.7. Natural Places

The seventh parameter is Natural Places. The keywords of this parameter are Caucasus, Mountains, European, Nature, Paradise, Traveler, Touches, Mix, Dream, Wonderful, and Enjoy. Examples of the tweets regarding this parameter are below:

"تمتع بجمال الطبيعة في جبال القوقاز الخلابة بعد الغداء في مطاعم الساحات الاوروبية والعالمية . . . "

*"Enjoy the beauty in spectacular Caucasus mountains after lunch in the restaurants of European and international squares . . . "*

"زور عسير واستمتع باستكشاف الطبيعة الساحرة في جبال السودة حتى قرية رجال ألمع التاريخية"

*"visit Asir and enjoy exploring the enchanting nature in the Souda Mountains to the historic village of Rijal Alma"*

"تروجينا . . . وجهة سياحية عالمية لعشاق الطبيعة والمغامرة والترفيه في أعالي جبال نيوم"

*"Trogena . . . a global tourist destination for lovers of nature, adventure and entertainment in the high mountains of NEOM"*

### 5.3.8. Seasonal Festivals

The eighth parameter is Seasonal Festivals. It is created by merging three clusters, 11, 16, and 17. The parameter captures various discussions about tourism related to seasonal entertainment festivals and events in Saudi Arabia. The keywords for the parameter include Tickets, Booking, Winterland, Season, Jeddah, Riyadh, Boulevard, Events, Jungle, Party, Music, Horrible, Fun, Show, and For Sale. For instance, the tweet outlined below was found in our dataset related to this parameter. It was posted by the booking company. This and similar tweets describe the Riyadh Season events and activities such as booking tickets and searching for flights to travel to Riyadh:

"احجز تذكرتك كافة انواع وفئات فعاليات موسم الرياض تذاكر حفلات ورحلات مناسبات والعاب ومنزهات في البوليفارد . . . "

*"Book your ticket for all types and categories of Riyadh Season events, tickets for parties, trips, events, games and parks on the Boulevard . . . "*

### 5.3.9. Temporal Progression (Tourist Attractions)

Figure 22 plots the temporal progression of the macro-parameter Tourist Attractions. We generated the plots using the topic-over-time method in the BERTopic library. The graph's horizontal line represents the timeline of the discovered parameters during the data period of March 2021 to October 2022, while the vertical line represents the number of tweets, which is referred to as the intensity. The tweets related to popular tourist national and international destinations and some tourism-related development projects in Saudi Arabia. The National Destination parameter was the most discussed topic in 2022.

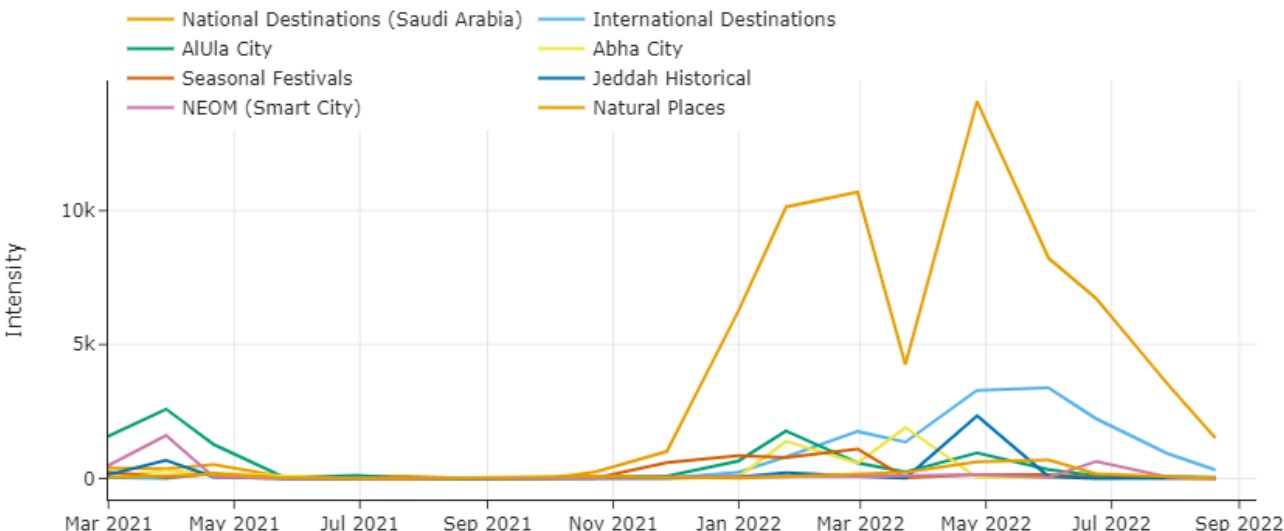

**Figure 22.** Temporal progression (macro-parameter: Tourist Attractions) (data source: Twitter).

### 5.4. Tourism Services

We discuss now the second macro-parameter, Tourism Services. It consists of five parameters. These are Programs, Tours, and Packages; Offers, Discounts, and Gifts; Tourist Guides; Restaurants and Hotels; and Medical Insurance and Internet Services.

#### 5.4.1. Programs, Tours, and Packages

The first parameter in this macro-parameter (ninth overall) is Programs, Tours, and Packages. This parameter is created by merging three clusters, numbers 5, 15, and 27. Most tourists, when planning their trips, look for reasonable and suitable tourism programs. Usually, tourism and travel companies offer a variety of tourism programs. This parameter covers these issues and is represented by the keywords Tourist, Programs, Most Suitable, Hotel Reservation, Rental, Distinguished, Tours, Reservations, Apartments, Cars, Offers, and Families. The following tweets provide examples of the posts:

<div dir="rtl">

"[شركةسياحية] يقدم لك أفضل البرامج السياحية في منصة وحدة من قبل شركات تنظيم الرحلات"

</div>

*"[Travel Company] offers you the best tourism programs in a single platform by tour operators".*

<div dir="rtl">

"عيش المغامرة الان في جورجيا مع برامج سياحية مختلفة من [شركة سياحية] للسياحة والسفر"

</div>

*"Live the adventure now in Georgia with different tourism programs from [Travel Company] for travel and tourism"*

<div dir="rtl">

"وزارة الثقافة بالتعاون مع وزارة السياحة ينظمون لزوار الحرمين برامج سياحيه ثقافيه مصممه للمعتمرين زيارات، لاماكن دينيه رحلات لطريق لهجرة. زيارات لمواقع الغزوات"

</div>

*"The Ministry of Culture, in cooperation with the Ministry of Tourism, organizes cultural tourism programs for visitors to . . . "*

<div dir="rtl">

"السياحه في . . . برامجنا تشمل شقق فاخره للعوائل والشباب استقبال وتوديع المطار فنادق ومنتجعات سيارة وسائق خاص رحلات يومي"

</div>

*"Tourism in . . . Our programs include luxury apartments for families and youth Airport reception and farewell Hotels and resorts Car and private driver Daily trips"*

The keywords related to Packages (Cluster 15) detected by our model include Camping, Accompany, Delivery, Reservation, Driver, Apartments, Guide, Airport, Prices, Tourists, Car, Flights, and Hotels. The following tweets provide examples of the posts:

<div dir="rtl">

"السياحه في . . . برامجنا تشمل شقق فاخره للعوائل والشباب استقبال وتوديع المطار فنادق ومنتجعات سيارة وسائق خاص رحلات يومي"

</div>

*"Tourism in . . . Our programs include luxury apartments for families and youth, Airport reception and farewell, Hotels and resorts, Car and private driver, daily trips"*

Some examples of tweets related to Tours or trips are provided below.

"رحلات سياحية لمزارع إنتاج الورد في عروس المصائف الطائف في صيف_السعودية"

*"Tourist trips to the rose production farms in the bride of summer resorts Taif City"*

"هنوفر لك حجز فندقي أو شقق فندقية و سيارة خاصة و رحلات سياحية"

*"We will provide you with a hotel reservation or hotel apartments, a private car and tourist trips"*

"قريباً رحلات كروز السعودية تنطلق مجدداً من جدة إلى 3 وجهات في البحر الأحمر السعودية ومصر والأردن"

*"Coming soon Saudi Cruise trips, it departs from Jeddah to three destinations in the Red Sea, Saudi Arabia Egypt Jordan"*

### 5.4.2. Offers, Discounts, and Gifts

The next parameter is Offers, Discounts, and Gifts. This parameter is created by merging three clusters, numbers 29, 2, and 21. The keywords related to this parameter (in all clusters) are Offers, Travel, Book, Enjoy, Save, Prices, Qatar, Resort, Beauty, Tourism, Services, UAE, Azerbaijan, Special, Discounts, Companies, Entrepreneurship, Gifts, Big, Inquiries, Winter, Best, Electronic, Lovers, and Location. The following tweets provide examples of posts related to this parameter:

"الان أفضل عروض السفر و السياحة إلى جزيرة ...الجميلة سارع بالحجز الان على افضل عروض السياحه وسفر..."

*"Now the best travel and tourism offers to the beautiful island . . . , hurry up to book now for the best tourism and . . . travel"*

"سياحة في جورجيا خصومات خاصة للأعداد الكبيرة وخصومات خاصة للشركات وهدايا للعرسان الجداد لأحلى شهر عسل"

*"Tourism in Georgia Special discounts for large numbers, special discounts for companies and gifts for newlyweds for the sweetest honeymoon"*

"خصومات وعروض ومبادرات خاصة بالمواطنين لتشجيع السياحة الداخلية"

*"Discounts, offers and initiatives for citizens to encourage domestic tourism"*

"استمتع باكتشاف جمال وجهات صيفالسعودية بخصومات حصرية تصل حتى 50%"

*"Enjoy discovering the beauty of Saudi summer destinations with exclusive discounts of up to 50%"*

"عروض_سياحية خصومات مميزة علي اسعار تذاكر الطيران علي احصل30%من سعر تذاكر سفرك..."

*"Tourist Offers Special discounts on airline ticket prices. Get a 30% discount on the price of your travel tickets . . . "*

### 5.4.3. Tourist Guides

The third parameter in this macro-parameter (eleventh overall) is Tourist Guides. It includes the following keywords: Inquiries, Guide, Your Trip, Offers, Reservations, Emirates, Bahrain, Oman, Tbilisi, Qatar, Tourism, Hotels, Areas, and Car. The following tweets provide examples of the posts:

"نسعد بخدمتكم في مجال الارشاد السياحي كمرشد سياحي في منطقة تبوك او نيوم اومحافظة العلا"

*"We are pleased to serve you . . . as a tourist guide in the Tabuk, Neom, or AlUla regions".*

"المرشد السياحي واجه مشرفه لبلاده فقد تم تأهيله في مهارات الارشاد السياحي والاسعافات الاولية ... بإشراف مباشر من وزارة السياحة ...فهو يقدم المعلومة الصحيحة عن المكان والزمان وخير معين في السفر"

*"The tour guide is qualified with the skills of tourist guidance and first aid . . . managed by the Ministry of Tourism . . . [tour guide] provides the correct information about the place and time and other information needed in travel"*

"سائق ومرشد سياحي في اندونيسيايتكلم الانجليزية والعربية"

*"Driver and tour guide in Indonesia who speaks English and Arabic"*

"رحلة سياحية لزيارة مرتفعات الشفا، تتضمن نشاط الهايكنج بمرافقة مرشد سياحي متخصص"

*"Tourist trip from Shefa Heights, hiking activity, accompanied by a specialized tour guide"*

### 5.4.4. Restaurants and Hotels

Restaurants and Hotels constitute the fourth parameter (overall twelfth) for the macro parameter Tourism Services. This parameter is created by merging three clusters, numbers 23, 24, and 7. It highlights the issues and importance of restaurants and accommodation services in tourism. The keywords include Places, Hotels, Restaurants, Resorts, Tourist, Nearest, Finest, Offer, Country, Vacation, Sites, Cities, Visitor, Family, Winter, Bahrain, and UAE. Following are example tweets:

"في نادي_جدة_لليخوت استمتعوا بتجربة ممتعة مع المطاعم المطلة على البحر"

*"In Jeddah yacht club, have an enjoyable experience with restaurants overlooking the sea"*

"أفضل اطلالات وأفخم الفنادق في جورجيا بأسعار خاصة"

*"The best views and the most luxurious hotels in Georgia at special prices"*

"لتعزيز قطاع السياحة وتطوير وجهات سياحية نوعية، وقع صندوق التنمية السياحي مع مجموعة فنادق ومنتجعات... مذكرة تفاهم لشراكة استراتيجية تطويريه لجذب الاستثمار السياحي بالمملكة"

*"To promote the tourism sector and develop quality tourist destinations, Tourism Development Fund signed up with a group of hotels and resorts a memorandum of understanding for a strategic development partnership to attract tourism investments in the Kingdom"*

"تُقدّم العلا مجموعة متنوعة من أماكن الإقامة والمنتجعات الراقية"

*"AlUla offers a variety of high-end accommodation and resorts"*

"تمتع بأجواء الاقامة في لأفخم المنتجعات والفنادق والأكواخ في جورجيا"

*"Enjoy the atmosphere of accommodation in the most luxurious resorts, hotels, and cottages in Georgia"*

### 5.4.5. Medical Insurance and Internet Services (MI2S)

The thirteenth parameter is Medical Insurance and Internet Services (MI2S). The global tourism industry has faced significant difficulties because of the COVID-19 pandemic. The requirements for traveling during and after the pandemic have been altered by policies and travel restrictions established in numerous countries. Travel insurance is important to protect from potential health hazards and financial losses. The keywords related to this parameter are Insurance, Get, Trip, SIM Cards, Internet, Driver, Taxi, Published, Tourist, Company, Best, Insurance, and Travel. The reason for the two topics—insurance and Internet services—is some companies offer the two services together.

"احجز رحلتك معنا وأحصل على تأمين السفر وشرائح انترنت مجاناوخصم خاص للحجز المسبق..."

*"Book your trip with us and get travel insurance, free internet SIM cards, and a special discount for pre-booking for reservations. . ."*

"الخطوط السعودية تشترط تأمين طبي للأطفال"

*"Saudi Airlines requires medical insurance for children"*

The following tweet was posted by Saudi Tourism Authority indicate the importance of obtaining medical insurance to cover COVID-19 risk when arriving at Saudi airports.

"يتعين على كافة الزوار الحصول على تأمين طبي ضد مخاطر كورونا؛ والتي يمكن الحصول عليها من جميع المطارات الدولية في المملكة عند الوصول"

*"All visitors are required to have medical insurance against corona risks, which can be obtained from all international airports in the Kingdom upon arrival"*

### 5.4.6. Temporal Progression (Tourism Services)

The temporal progression of the macro-parameter Tourism Services, which includes five parameters, is shown in Figure 23. A mix of behaviour can be observed for all parameters. Overall, the Offers, Discounts, and Gifts parameter shows the highest intensity among all, and this could be due to businesses posting tweets about their offers.

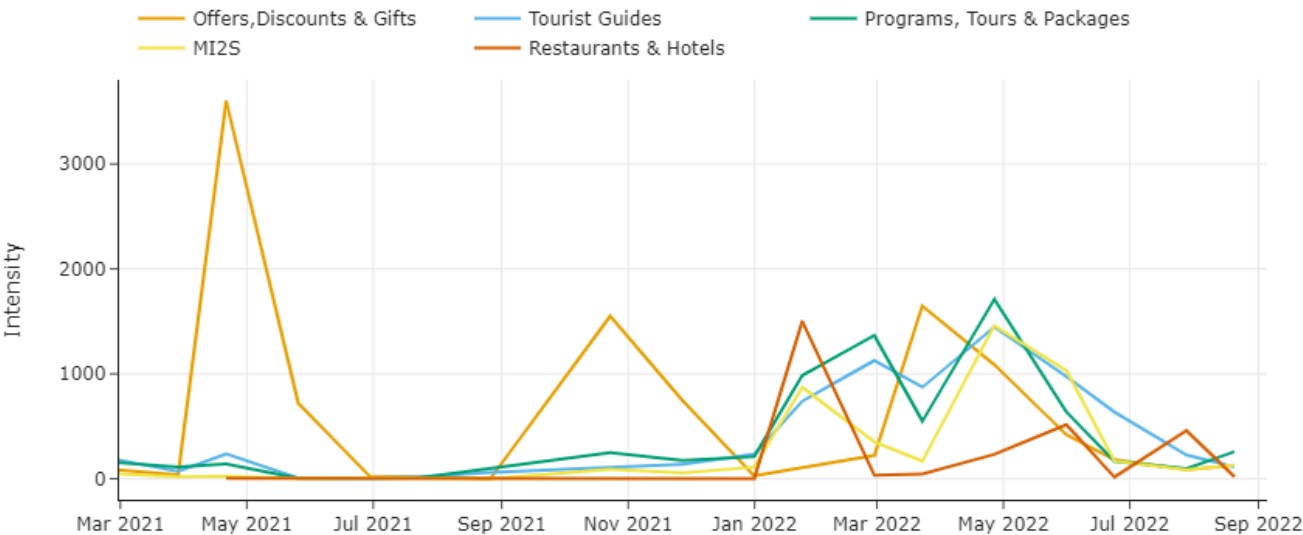

**Figure 23.** Temporal progression (macro-parameter: Tourism Services) (data source: Twitter).

## 6. Discussion

The aim of this paper is to gain a comprehensive understanding of tourism and develop a theory and approach for smarter, sustainable tourism through the use of cutting-edge technologies. The study modelled the tourism industry using scientific research papers and tweets. It used a data-driven approach with deep learning and big data to extract parameters from both academic literature and public opinions on Twitter to give a comprehensive view of the industry from two different perspectives. A software tool for smart tourism was developed with four modules using BERT embeddings, UMAP, HDBSCAN, and TF-IDF, grouping discovered parameters into macro-parameters, validated internally and externally, and visualized with Seaborn, Plotly, and Matplotlib libraries. The paper also presented a comprehensive knowledge structure and literature review of the tourism sector, drawing on more than 250 research articles.

The academic-view dataset was constructed using 156,759 English research articles from the Scopus database covering 2000–2022. It was used to uncover key elements of academia-oriented tourism. Thirty-three parameters related to tourism were identified and grouped into four categories: Tourism Types, Tourism Planning, Tourism Challenges, and Media and Technologies. The "Tourism Types" macro-parameter discovered that the tourism industry encompasses a variety of activities and experiences such as urban, rural, beach and marine, national parks, wildlife, adventure, diving, sports, space, culture and heritage, music and art, and religious tourism. Activities range from city sightseeing, rural life experience, swimming, snorkeling, safari, adventure, visiting museums and historical

sites, attending concerts, visiting art galleries, and religious tourism. The "Tourism Planning" macro-parameter captured details of planning for tourism that involves investing in education and training, utilizing air transport, and considering the hotel and food service industries to drive sustainable economic growth in communities. Forecasting methods are used to predict future demand. It also helps to develop underdeveloped communities by providing jobs and income opportunities.

The "Tourism Challenges" parameter captured several challenges that the tourism industry is facing such as climate change, poor air and water quality, natural disasters, and pandemics. Climate change is affecting the availability of tourist destinations and activities. Poor air and water quality can make tourist destinations less attractive and have negative effects on visitors' health. Natural disasters and pandemics can damage tourism infrastructure and disrupt travel plans. To overcome these challenges, research in sustainable tourism is underway such as reducing carbon emissions, improving waste management, and promoting sustainable transportation options, which will minimize negative impacts on the environment and local communities while promoting the long-term viability of the tourism industry. The "Media & Technology" parameter captured the major impact that media and technology are having on the tourism industry. Film and TV have been used traditionally to showcase destinations. Online and social media platforms have now become essential for promotion. Mobile apps have made it easier for travelers to plan and book trips, and virtual reality technology allows people to experience a destination before visiting or enhance their experiences during visits.

The Twitter dataset for this study was collected for 18 months from March 2021 to August 2022 and consisted of 485,813 tweets related to the public perception of tourism in Saudi Arabia. The dataset was limited to the region to focus on local tourism issues and was modelled to reveal 13 parameters, grouped into two broader categories: Tourist Attractions and Tourism Services.

The "Tourist Attractions" parameter captured a diverse range of tourist attractions that Saudi Arabia offers for both domestic and international travelers, including popular National destinations such as the NEOM Smart City, AlUla City, Abha City, and Jeddah, and many natural places such as deserts, mountains, and beaches. Additionally, Saudi Arabia hosts many seasonal festivals that allow tourists to experience the country's culture and traditions. With a variety of options catering to different interests, Saudi Arabia is an attractive destination for tourists. The parameter also captured many international destinations that are popular for Saudi tourists including, Qabil, Qubam and Baku (Azerbaijan), Batumi, Tbilisi, and Borjomi (Georgia), and Dubai, Emirates, Qatar, Bahrain, Bosnia, and Trabzon (Tukiye). The "Tourism Services" captured a wide range of tourism services Saudi Arabia offers for both domestic and international travelers, including tour packages, discounts and gifts, and tourist guides. They also offer a variety of hotels and restaurants and options for medical insurance and internet services.

The two perspectives, academic versus public, or international versus national, are not isolated and have some impact on each other, but they still have distinct and significant variations. In Figure 1, in comparing these two perspectives, we presented a multi-perspective taxonomy of the tourism sector, combining academic, public, international, and national/cultural (Saudi Arabia) perspectives. It offered a holistic understanding of the industry, including 15 types of tourism, various planning dimensions, major challenges, and the impact of media and technology (academic and international view). The national/public perspective in Saudi Arabia focuses on tourist attractions and services such as medical insurance, including recent developments such as the NEOM smart city, AlUla city, and seasonal festivals.

Figure 24 presents a data-driven framework for smarter tourism, aimed at improving tourist experiences and promoting sustainable practices. The framework is based on data sources such as social media, academic articles, government, and industry, among others. The objectives include improving the quality of services, cultural sustainability, economic sustainability, environmental sustainability, and tourist experiences. Enablers

for achieving these objectives include tourism satisfaction, efficiency, experiences, and emerging technologies. The framework also lists challenges in tourism, such as climate change, air pollution, water quality, security, lack of awareness, and food safety. These challenges should be addressed by the framework to achieve the desired goals.

**Data-Driven Framework for Smarter and Sustainable Tourism**

**Tourism Data Sources**

| Social Media | Scientific Literature | Government & Industry |

**Challenges**

**Objectives**

**Factors, Solutions, & Enabling Technologies**

Tourism Satisfaction (Hospitality, Services, Responsiveness, Cultural Heritage, Events & Activities)

Tourism Efficiency (Planning, Economic Growth, Education, Infrastructures, Awareness, etc.)

Tourist Experiences (Travel Services, Attractions, Motivations, etc.)

Emerging Technologies (Mobile Applications, Social Media, Virtual Reality, TV Media)

Sustainability (Environment Friendly, Population, Green Hotels, Health, etc.)

Challenges:
- Climate Change
- Air Pollution
- Water Quality
- Poor Communities
- Disasters & Pandemics
- Overtourism
- Safety & Security
- Economic Impact
- Lack of Infrastructures
- Lack of Awareness
- Food Safety

Objectives:
- Quality of Service
- Culture Sustainability
- Environment Sustainability
- Economic Sustainability
- Social Sustainability
- Service Infrastructure
- Attractions & Promotions
- Tourist Experiences
- Productivity, Efficiency & Robustness

**Figure 24.** Data-driven framework for smarter sustainable tourism.

*6.1. Novelty and Utilization*

The tourism industry is constantly evolving and facing new challenges such as emerging technologies, global conflicts, energy and monetary crises, pandemics, and disasters. A literature review (see Section 2) and an extensive analysis of current research (see Section 4) show that the field is fragmented and narrowly focused. To effectively navigate and improve this dynamic sector, a holistic approach is needed to study tourism. A holistic view of the tourism industry considers all aspects of the industry, including economic, social, environmental, and cultural factors. By taking this comprehensive approach, decision-makers can gain a deeper understanding of the interconnectedness of different aspects of tourism and how they impact one another. This knowledge can be used to drive sustainable tourism practices and destination development and enhance visitor experiences. Additionally, a holistic approach can help identify potential challenges and opportunities in advance, allowing for proactive rather than reactive solutions. This proactive approach is crucial in today's fast-paced environment as it helps to stay ahead of industry trends and mitigate potential negative impacts. Furthermore, a holistic approach to tourism can have a positive impact on both tourists and local communities, promoting sustainable tourism practices that benefit the environment and local communities and creating better visitor experiences that lead to repeat visits and positive word of mouth.

Moreover, the relationship and effects of tourism on local cultures in the emerging digital world have also attracted limited attention, particularly in Saudi Arabia. In these contexts, research on big data analytics of social and digital media, particularly in the Arabic language, is limited. Saudi Arabia is a rapidly developing country with a unique culture and a diverse range of tourist attractions. Conducting tourism research in this country can provide insight into the latest trends and developments in the industry and

help to identify potential opportunities for tourism growth. It can also allow researchers to explore the country's attractions and learn more about its culture and history.

This paper presents an approach for leveraging machine and deep learning to gather holistic, multi-perspective (e.g., local, cultural, national, and international), and objective information on any subject including tourism. By providing powerful tools and resources to analyze various datasets, it makes it possible to uncover crucial information on matters of public, academic, industrial, and government interest. The research and insights discovered in this paper make a significant contribution to our understanding of the tourism industry and have the potential to shape public perception, guide future research, and inform decision-making by the public, the government, and other stakeholders.

The proposed approach not only enhances the theory and practice of AI-based methods for information discovery but also extends the use of the scientific literature, Twitter, and other media and data sources for information and parameter discovery to enable autonomous capabilities for holistic and dynamic optimizations in everyday applications, systems, and platforms. Furthermore, it promotes novel approaches to research in the tourism sector using the information discovery approach, ultimately giving rise to the development of smart and sustainable societies, economies, and the planet, which is a paramount concern for today's world.

As technology advances, more and more systems such as self-driving cars, web services, drones, and robots in manufacturing and farming are becoming autonomous. This trend is likely to continue, and we will see this kind of autonomous functionality being incorporated into an even wider variety of systems, including those used in industry, city, and country management. Even when a system is not fully autonomous, understanding its parameters is still important, as they form the basis for decision-making and problem-solving in the design and operation of the system.

## 7. Conclusions

The fragility of the tourism industry, which prior to the COVID-19 pandemic contributed 10.3% to the global GDP and employed 333 million people, is being exposed by global natural and manmade events and requires collaboration and comprehensive understanding for responsible and innovative growth towards sustainable and smart tourism. The aim of this study is to gain a comprehensive understanding of tourism, drive future research through cutting-edge technologies (artificial intelligence, big data, and others), and ultimately develop a theory and approach for smarter tourism that supports sustainable future societies. This paper presented a machine learning approach to extract parameters from the academic literature and public opinions on Twitter to provide a holistic view of the tourism industry and promote sustainable and smart tourism through improved AI-based information discovery. The approach modelled 156,759 research articles and 485,813 tweets to identify 33 distinct parameters in 4 categories for the academic perspective and 13 parameters for public perception. The paper presents a comprehensive knowledge structure and literature review of the tourism sector, drawing on more than 250 research articles.

### 7.1. Theoretical and Practical Implications

The work presented in this paper has significant theoretical and practical implications for the tourism industry and beyond and is of critical importance in the current rapidly evolving technological landscape (see also Section 6.1). By highlighting the need for a comprehensive, holistic approach to studying the tourism industry, the paper provides a framework that can be used to navigate the complex challenges facing the industry, such as emerging technologies, global conflicts, pandemics, and disasters.

The practical implications of this work are numerous and far-reaching, with the development of powerful tools and resources for analyzing diverse datasets representing a major step forward in the effective management and optimization of complex systems. These tools and resources are critical in today's fast-paced environment, where proactive

rather than reactive solutions are needed to stay ahead of industry trends and mitigate potential negative impacts.

Additionally, the proposed approach has the potential to extend the use of scientific literature, Twitter, and other media and data sources for information and parameter discovery to enable autonomous capabilities for holistic and dynamic optimizations in everyday applications, systems, and platforms. By leveraging machine and deep learning to gather objective and multi-perspective information, decision-makers can make more informed choices, leading to the development of smart and sustainable societies, economies, and the planet.

Overall, this work represents a significant contribution to our understanding of the tourism industry and has the potential to shape public perception, guide future research, and inform decision-making by the public, government, and other stakeholders. It is essential reading for anyone interested in the future of the tourism industry and the development of sustainable, technology-driven solutions to the challenges facing our planet.

### 7.2. Limitations

While the work presented in this paper is undoubtedly important and valuable, it is important to acknowledge its limitations. One of the primary limitations is that the study focuses on the tourism industry in Saudi Arabia, and while this is a rapidly developing country with unique cultural and tourist attractions, the findings may not be applicable to other countries or regions.

Additionally, the proposed approach for leveraging machine and deep learning to gather objective and multi-perspective information has its own limitations. For example, the accuracy and reliability of the data sources used to train the machine learning algorithms can affect the quality of the insights generated. Similarly, the complexity of the algorithms used can make it difficult to interpret the results and identify potential biases.

Another limitation is that the approach proposed in this paper is heavily reliant on technological infrastructure, which may not be available or accessible in all regions. This could limit the applicability of the approach in certain contexts and may prevent some decision-makers from accessing the insights and resources that it provides.

### 7.3. Future Work

The paper belongs to our extensive research on utilizing Information and Communication Technology (ICT) to tackle issues in smart cities and societies. Our work encompasses deep journalism [51,261], labor economics [262], transportation [51], smart families and homes [52], healthcare [62,263], education during COVID-19 [64], and event detection [67]. In the future, we aim to enhance the methodology in this paper through advanced deep learning techniques and apply them to enhancing tourism and other societal, economic, environmental, and cultural issues. This research utilized Scopus database and Twitter data to uncover parameters. In the future, we plan to integrate other scientific databases, social and digital media, and additional data sources to expand the scope of our findings and provide a more comprehensive understanding. Finally, we note that the development of sustainable tourism must necessarily be based on intangible factors of territorial development, such as intellectual capital in particular, such as noted in [264,265]. Future work will also look into this direction.

**Author Contributions:** Conceptualization, R.A. and R.M.; methodology, R.A. and R.M.; software, R.A.; validation, R.A. and R.M.; formal analysis, R.A., R.M. and A.A.; investigation, R.A., R.M. and A.A.; resources, R.M. and A.A.; data curation, R.A.; writing—original draft preparation, R.A. and R.M.; writing—review and editing, R.M. and A.A.; visualization, R.A.; supervision, R.M. and A.A.; project administration, R.M.; funding acquisition, R.M. All authors have read and agreed to the published version of the manuscript.

**Funding:** The authors acknowledge with thanks the technical and financial support from the Deanship of Scientific Research (DSR) at the King Abdulaziz University (KAU), Jeddah, Saudi Arabia, under Grant No. RG-11-611-40.

**Institutional Review Board Statement:** Not applicable.

**Informed Consent Statement:** Not applicable.

**Data Availability Statement:** The data used in this paper can be made available subject to the data providers' terms and conditions.

**Acknowledgments:** The work carried out in this paper is supported by the HPC Center at the King Abdulaziz University. The training and software development work reported in this paper was carried out on the Aziz supercomputer.

**Conflicts of Interest:** The authors declare no conflict of interest.

## Abbreviations

| | |
|---|---|
| BERT | Bidirectional Encoder Representations from Transformer |
| HDBSCAN | Hierarchical Density-Based Spatial Clustering of Applications with Noise |
| UMPA | Uniform Manifold Approximation and Projection |
| TF–IDF | Term Frequency-Inverse Document Frequency |
| ICTs | Information and Communication Technologies |
| UNWTO | World Tourism Organization |
| AI | Artificial intelligence |
| NLTK | Natural Language Toolkit |
| NLP | Natural language processing |
| EGUC | Economic Growth for Underdeveloped Communities |
| WQPM | Water Quality & Pollution Management |
| MI2S | Medical Insurance & Internet Services |

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
