# Peer review of "Smarter Sustainable Tourism: Data-Driven Multi-Perspective Parameter Discovery for Autonomous Design and Operations"

_sustainability, doi:10.3390/su15054166_

Round 1

Reviewer 1 Report

Thank you for allowing me to review this manuscript. I find the current manuscript interesting and enjoy reading it. However, it was one of the longest manuscripts I reviewed. I have a few suggestions to improve the manuscript:

Abstract

The practical and theoretical implications of the study or recommendation should be summarized in a few sentences and included in the abstract.

Introduction

Please consider deleting p.1 lines 38-42 and start the Introduction with Title 1.1.

Please consider changing the sub-title 1.5 and 2.

1.4 Starting from the first sentences and continuing until the end of the manuscript, several sentences need in-text references or citations (e.g., page 3, lines 144; 146 & p.4, lines 149-154 etc.).

I like the last section 2.3 (good job).

Literature Review/ Methods/ Results:

There were no concerns in those sections.

Conclusion:

The authors should summarized the goal of the study and the key findings. However, they should ensure the ‘so what’ question is answered – explain the broader implications or significance of the study. As a final suggestion, they should discuss the studys' limitations and implications of the study. I would suggest the author(s) improve the conclusion section and create implications and limitations and future study suggestion sections. Overall, the study's contribution to theory and practice is not clear, noticeable, or evident. I wish the author the best of luck with the revision.

Reviewer 2 Report

This research is interesting but has some flaws:

1) "It is high time for investors and other stakeholders in the tourism sector in Saudi 76 Arabia." Why? Authors shall explain why.

2) The introduction section seems like a project report rather than a paper. Authors are suggested to revise this section.

3) The main theme of paper is smarter sustainable tourism, which is introduced in section 1.4. Authors are suggested to explain what and why smarter sustainable tourism in the first half of the introduction.

4)Methods and alaysis seems okay.

5) ". None of the earlier works 303 have used a methodology and accordingly developed a tool that uses several cutting-edge 304 deep learning and big data methods for analysis, discovery, and visualisation of information and knowledge for policymaking, design, and operations". The authors claimed that the methodology employed is new to the tourism context. Authors are suggested to cross-check their claim.

6) This manuscript seems to me like a project report. Authors are suggested to reduce the word count because it is hard for Sustainability readers to have a continuance focus.

Reviewer 3 Report

The study deals with a very interesting topic, which is currently widely discussed in the academic community. Examining the possibilities of using AI-based tools contributes to solve numerous problems in the field of sustainable tourism development management. However, I would like to make some remarks. Some claims especially in the introduction need to be supported by evidence on research already conducted. For example, lines 149-150, and 158-160 there is a study that should be referenced in this case e.g. Matlovicova, K; et al. 2016. Environment of estates and crime prevention through urban environment formation and modification. Geographica Pannonica 20/3, 168-180. In the case of NEOM city (rows 109) it is a sustainable development project based on the concept of the so-called 15-minute city and should refer to, for example, the study Mocak, P; et.al. 2022. 15-Minute City Concept as a Sustainable Urban Development Alternative: A Brief Outline of Conceptual Frameworks and Slovak Cities as a Case. Folia Geographica, 64 (1) , pp.69-89. Undoubtedly, support through improved information discovery based on artificial intelligence is beneficial (row 1861),but the development of sustainable tourism must necessarily be based on intangible factors of territorial development, such as intellectual capital in particular. This assertion that intellectual capital is the basis for the development of countries and regions with the help of smart tools based on artificial intelligence is confirmed, for example, by studies: Pachura, P., et al. 2018. Identification of Intellectual Capital Performance Using Data Envelopment Analysis. Advances in Spatial Science, 115-130 and also Neumannová M. 2022. Smart Districts: New Phenomenon in Sustainable Urban Development. Case Study of Špitálka in Brno, Czech Republic, Folia Geographica 64/2, pp. 27-48. It is a quality study based on a well-developed and original methodology for obtaining and processing relevant data. The paper has a logical structure, relies on relevant sources and provides an interesting perspective on the undoubtedly widely discussed problem the utilization of artificial intelligence tools in sustainable tourism development. The above comments in no way diminish the quality of the study. It is balanced in content, uses correct methods and I definitely recommend it for publication after minor changes.

Reviewer 4 Report

-
